# Language Adaptation on a Tight Academic Compute Budget: Tokenizer Swapping Works and Pure `bfloat16` Is Enough

Konstantin Dobler [1]   Gerard de Melo [1]

## Abstract

We investigate continued pretraining of LLMs for language adaptation on a *tight academic budget*: a setting in which only a few GPUs can be used in parallel, for a heavily constrained duration. We focus on adapting Mistral-7B to German or Arabic and evaluate several techniques to improve efficiency and effectiveness in this setting. Our German models adapted on this tight compute budget underperform compared to the base Mistral-7B, while our Arabic models outperform several baselines, showing that for sufficiently well-represented languages, continued pretraining for specialization is not always helpful. Our main findings focus on training precision and tokenizer swapping. Our results show that pure `bfloat16` training is a viable alternative to mixed-precision training, while being much faster when only using a few GPUs. Swapping the tokenizer for a specialized one yields more efficient tokenization and is competitive with the original tokenizer, which already contains some German tokens, but did not significantly increase performance for German. Code and model weights are available on GitHub.

## 1. Introduction

Large language models (LLMs) can be incredibly useful tools – if they handle your language well. Unfortunately, for people who speak languages that are not among the small subset of high-resource languages, this is often not the case. One remedy is to adapt an existing LLM to your desired target language via continued pretraining (Chau et al., 2020; Gururangan et al., 2020). This leverages the enormous amount of compute spent on training the base LLM. Still, as models have gotten bigger, even the compute required for

continued pretraining can be too much for many academic labs to handle, requiring the support of large governmental or industrial compute grants.

In this work, we use the term **"tight academic (compute) budgets"**, referring to constrained access to a limited number of GPUs (such as only two or four GPUs), constrained GPU memory capacity (e.g., 40GB vs. 80GB Nvidia A100 GPUs), and constrained access durations.[1] As academic labs commonly do have access to server-grade rather than consumer-grade GPUs, our definition does encompass the availability of (a few) server-grade GPUs, such as Nvidia A100s, in this setting. We investigate language adaptation of LLMs on such a tight academic budget. We start with the training recipe of the recent LLM adaptation project LeoLM (Plüster et al., 2023) adapting Mistral-7B[2] (Jiang et al., 2023a) to German, and modify it for our tight academic compute budget setting. Selecting German as the target language allows us to draw a rough lower bound on the viability of language adaptation on a tight academic budget since Mistral-7B has already seen German during pretraining. For unseen languages, the benefits of language adaptation will be more pronounced. We verify this in an additional "hindsight study" targeting Arabic after our main experiments in German. The focus of our study *is not* to produce a new state-of-the-art language model in German or Arabic. Rather, we investigate the viability of efficient training techniques to inform future language adaptation attempts on a tight compute budget.

We specifically focus on two dimensions of efficiency: tokenizer swapping and training precision. Firstly, we investigate the necessity of the commonly used mixed-precision training (Micikevicius et al., 2017) over training in pure `bfloat16` precision. In settings with only two or four GPUs being used in parallel for adapting 7 billion parameter models, mixed-precision `bfloat16` training will run out of memory or is only possible if used with inefficient memory-saving techniques like activation checkpointing.

---

[1]Hasso Plattner Institute / University of Potsdam, ELLIS Unit Potsdam, Germany. Correspondence to: Konstantin Dobler <konstantin.dobler@hpi.de>.

Accepted to the Workshop on Advancing Neural Network Training at International Conference on Machine Learning (WANT@ICML 2024).

---

[1]Typically, unless one is willing to exchange significant social goodwill into compute by delaying the work of all other people in the lab, multiple GPUs can be blocked for only a short number of days (e.g., over weekends or holidays).

[2]In this work, Mistral-7B specifically refers to `mistralai/Mistral-7B-v0.1`.

This is avoided in pure `bfloat16` training by eliminating the need to store full-precision optimizer states and a model weight copy, as required for mixed precision.

Secondly, we investigate swapping out Mistral-7B's original tokenizer with a specialized German tokenizer. Many recent language adaptation studies choose to retain the base LLM's original tokenizer (e.g., Kuulmets et al., 2024; Huang et al., 2023; Plüster et al., 2023). Others opt to extend the original vocabulary rather than replacing it (Zhao et al., 2024b; Fujii et al., 2024; Nguyen et al., 2023). A more specialized tokenizer has better fertility on our training data, allowing us to train on a greater total number of words (although not tokens) for the same compute and offers a semantically (more) sensible tokenization of text. Swapping instead of extending the tokenizer, resulting in smaller embedding matrices, further boosts training efficiency due to the faster unembedding matrix multiplication and softmax operations and a smaller memory footprint.

We summarize our results as follows:

- We recommend pure `bfloat16` training with small caveats (see Section 4.1) for its large training efficiency gains, especially in tight academic compute budget settings with fewer than eight parallel GPUs, challenging the commonly used mixed-precision `bfloat16` paradigm for continued training with extensive analysis and experiments.

- We find tokenizer swapping performs on par with keeping the original tokenizer even on a small compute budget, although it does not improve task performance (see Section 4.2), while yielding more efficient tokenization.

- Continued pretraining of Mistral-7B on German decreased German task performance, whereas adapting to Arabic led to significant gains. This demonstrates that language adaptation is not always beneficial when the target language is already well-represented (see Section 4.3).

We further provide an introduction to mixed-precision LLM training in Section 2. We provide more details on our experimental setup in Section 3 and present the results in Section 4. We discuss related work in Section 5 and the limitations of this study in Section 6. We publish our code and model checkpoints under https://github.com/konstantinjdobler/tight-budget-llm-adaptation.

## 2. Background: Precision Types

One of our main research questions is whether mixed-precision training is necessary for continued pretraining.

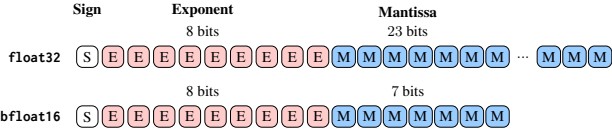

Figure 1: Illustration of the memory layout of `float32` and `bfloat16`, based on García-Nava et al. (2022).

In this section, we provide a short introduction to mixed-precision `bfloat16` training. The educated reader is encouraged to skip ahead to Section 3.

Nowadays, the most common training setup is `bfloat16` mixed-precision training.[3] In mixed-precision training, the forward and backward pass is computed in `bfloat16` instead of `float32`, as this increases computational efficiency. However, `bfloat16` has less precision than `float32` due to its memory layout with fewer mantissa bits (as illustrated in Figure 1). Hence, a master copy of the model weights and the optimizer states are stored in `float32` and used during the optimizer step to offset this reduced precision, at the cost of increased memory usage.

In settings with constrained access to GPUs, such as our tight academic budget setting, this additional memory cost can force the use of smaller batch sizes and activation checkpointing, which result in significantly lower training efficiency. For example, in our experiments with two 80GB GPUs, pure `bfloat16` was 39% faster than mixed-precision `bfloat16` training. With access to only a single 80GB GPU, mixed precision proved impossible due to out-of-memory errors. Therefore, we extensively analyze the viability of pure `bfloat16` training for language adaptation in the tight academic compute budget setting.

**The Numerics of `bfloat16`.** Let us take a closer look at how floating point numbers are encoded. The formula for the actual value of a floating point number $n$ is:

$$n = s \times 2^{e-127} + s \times 2^{e-127} \times 2^{-m}$$

where $s$ is the sign ($-1$ or $1$), $e$ is the number encoded by the exponent bits, and $m$ is the number encoded by the mantissa bits. An intuitive model of this is that we have *base* numbers represented by the first term $s \times 2^{e-127}$ that are powers of two over the entire representable range of the datatype (e.g., $2, -2, ... 64, -64, ...$) and *"fractional"* numbers represented by the second term $s \times 2^{e-127} \times 2^{-m}$ that add a fraction of that power of two until the next power of two is reached. Note that we always have the same amount of fractions between any two powers of two. For `bfloat16`, we *always*

---

[3]Recently, the `fp8` type supported by Nvidia H100 GPUs has also gained popularity, but we leave this for future work. The problems we illustrate for `bfloat16` are expected to be exacerbated for `fp8`.

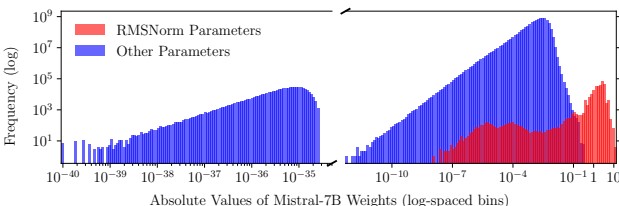

Figure 2: Histogram of absolute individual parameter weight values of Mistral-7B, separately highlighting RMS-Norm and non-RMSNorm weights.

have $128 = 2^7$ fractions, since we have 7 mantissa bits. Therefore, `bfloat16` will have very low precision for very large numbers. We have to cover the space between 256 and 512 with the same number of fractions as for the space between 2 and 4. This means that we can only represent 128 *floating point* numbers between 256 and 512, which only covers every second *integer*. All other numbers get rounded to the nearest representable one. The maximum $\epsilon$ so that $x + \epsilon$ still results in $x$ grows as $x$ grows, or – in other words – if $x$ is larger, the minimum representable change of $x$ is also larger.

**Large weights lead to problems when training in pure `bfloat16`.** When we have some small update $u$ that we want to add/subtract from a weight $W$, because of the (im-)precision of `bfloat16`, this update has no effect when $u < W/128$ ($128 = 2^7$ for the 7 mantissa bits). This is because $u$ is not large enough to push $W$ to the next fractional value that is representable in `bfloat16`. So how big of a problem is this in practice? Usually, our network's weights are not *that* big. If an individual parameter's value is $0.05$, then the smallest possible update that does not disappear due to `bfloat16` numerics is $0.05/128 \approx 0.00039$. As a side note: the same effect exists for training in `float32`, but only when $u < W/2^{23}$ for 23 mantissa bits in `float32`. Arguably, this is not a problem in practice.

**Case study: Mistral-7B.** In Figure 2, we plot the absolute values of Mistral-7B's weights. We see that most weight values are relatively small, which is good, as even very small update values will not disappear in pure `bfloat16` training. However, when considering the average values of RMSNorm weights and other weights separately, we observe that the RMSNorm weight values are much larger than other weights. This means that small updates to their weights during training in pure `bfloat16` are more likely to disappear. We evaluate and discuss this in Section 4.1.

## 3. Experimental Setup

We now describe the experimental setup of our main experiments and hindsight study.

| Hyperparameters | Ours (Main) | LeoLM |
|---|---|---|
| Training steps | 7,680 | 15,360 |
| Warmup steps | 76 | 153 |
| Batch size | 256 | 512 |
| Context length | 4,096 | 8,192 |
| Total training tokens | 8 billion | 64 billion |
| Optimizer | AdamW | AdamW |
| Learning rate | $4 \times 10^{-5}$ | $2 \times 10^{-5}$ |
| Learning rate decay | Cosine to $2 \times 10^{-6}$ | Cosine to $2 \times 10^{-6}$ |
| Adam $\beta$ | (0.9, 0.95) | (0.9, 0.95) |
| Weight decay | 0.05 | 0.05 |

Table 1: Hyperparameters for our main language adaptation experiments, with a comparison to the hyperparameters used by LeoLM (Plüster et al., 2023). The learning rate schedule is modified in our hindsight study, see Section 3.2.

### 3.1. Main Experiments

We build on the training recipe from LeoLM, which adapts Mistral-7B to German via full fine-tuning of all parameters. We report our hyperparameters in Table 1. In particular, we use a context length of 4,096 rather than 8,192 tokens to reduce the required GPU memory, as we cannot shard the model across a large number of GPUs. Additionally, we only train for half the number of optimizer steps and use a total batch size of 256 instead of 512 to fit our tight academic compute budget setting. These are our main adjustments to reduce the amount of total compute necessary to complete the training. As a result, our training recipe trains for 8 billion tokens compared to LeoLM's 64 billion tokens, which amounts to 12.5%.

The LeoLM recipe is intended to produce a bilingual model by mixing in English data during continued pretraining and using a lower learning rate, which has been shown to reduce catastrophic forgetting but decreases adaptation to the target domain (Ibrahim et al., 2024). In our work, we instead choose to focus solely on the target language. This allows us to use a higher learning rate, which has also been shown to help adaptation (Ibrahim et al., 2024). After initial probing experiments[4], we double the learning rate to $4 \times 10^{-5}$.

We use the data from the German split of OS-CAR23.01 (Abadji et al., 2022) and filter out all documents with quality warnings.[5] As noted, we do not mix in additional English data or code. For tokenizer swapping, we train a sentencepiece (Kudo & Richardson, 2018) BPE tokenizer with byte-fallback on a subset of our training data with a vocabulary size of 32,768 tokens for optimized Nvidia Tensor Core usage. Importantly for training sentencepiece

---

[4]We observe very large gradient norm spikes early on when training with $6 \times 10^{-5}$, $2 \times 10^{-4}$, and $3 \times 10^{-4}$.

[5]We remove samples with any of the following quality warnings: `adult`, `noisy`, `header`, `footer`, `tiny`, `short_sentences`. The quality warnings are provided by the OSCAR corpus.

tokenizers on web-crawled data, we use a character coverage of slightly less than $100\%$ to prevent very infrequent characters from being added to the vocabulary (Dobler & de Melo, 2023).

For downstream task evaluation, we rely on a suite of German benchmarks by Plüster (2023), which were also used for LeoLM. We use their fork of the `lm-eval-harness` (Gao et al., 2023) to evaluate our checkpoints. The individual datasets used are translated versions of MMLU (Hendrycks et al., 2021), TruthfulQA (Lin et al., 2021), ARC (Clark et al., 2018a), HellaSwag (Zellers et al., 2019), as well as PAWS-X (Yang et al., 2019) and LAMBADA-OpenAI (Paperno et al., 2016; Radford et al., 2019).

We perform the computation of this work in a real-world tight academic compute budget setting. We use either two or four Nvidia A100 80GB, A100 40GB, H100 80GB, or A6000 48GB GPUs for parallel training with PyTorch FSDP (Zhao et al., 2023). Depending on the number of available GPUs and training precision, we adjust efficiency settings, such as micro-batch size, activation checkpointing, or different ZeRO (Rajbhandari et al., 2019) stages of PyTorch FSDP as needed to fit the available GPU memory. We use the FlashAttention self-attention and RMSNorm CUDA kernels (Dao, 2023).

### 3.2. Hindsight Study

We conduct an additional *hindsight study* using pure `bfloat16` and tokenizer swapping based on the findings from our main experiments for further analysis. We additionally make the following changes to our original training recipe: (i) We use training data from CulturaX (Nguyen et al., 2024) instead of OSCAR23.01, as CulturaX has advanced cleaning and deduplication steps already applied. (ii) We additionally target Arabic alongside German to evaluate language adaptation when the target language has not had a significant share during the base model's pretraining. (iii) We integrate (continued) training improvements by using an infinite learning rate schedule following Ibrahim et al. (2024) and adjusted attention masks to prevent cross-document attention. We use the same hyperparameters as the main experiments except for the modified learning rate schedule, which uses a cosine decay for $60\%$ of steps after warmup, followed by a phase of a constant learning rate at $1.65 \times 10^{-5}$.

To explicitly study the effects of pure `bfloat16` during the low learning rate annealing phases of a learning rate schedule, we additionally train a version for both Arabic and German where we employ mixed-precision `bfloat16` just during the annealing phase (the last $14\%$ of training) by continuing from the pure `bfloat16` checkpoint before the start of the annealing phase. For German, we use the same

| Precision | GPUs | Best Config | GPU Hours | Speedup |
|---|---|---|---|---|
| mixed | 1 | OOM | OOM | – |
| pure | 1 | (1, no, N/A, N/A, paged) | 228.3 | ∞ |
| mixed | 2 | (4, yes, full, sync, paged) | 317.0 | – |
| pure | 2 | (1, no, grad_op, no_sync, no_paged) | 227.7 | 39.2% |
| mixed | 4 | (8, yes, full, sync, no_paged) | 295.7 | – |
| pure | 4 | (1, no, grad_op, no_sync, no_paged) | 225.8 | 31.0% |
| mixed | 8 | (8, yes, full, sync, paged) | 298.0 | – |
| pure | 8 | (1, no, grad_op, no_sync, no_paged) | 229.6 | 29.8% |

Table 2: Benchmark of total compute budget expended for each run configuration in H100 80GB GPU hours. We searched for the best (`micro-batch size`, `activation checkpointing`, `FSDP sharding`, `gradient syncing`, `paged AdamW`) tuple for each combination of precision and number of GPUs. OOM: Out-of-memory. More details are provided in Appendix E.

tokenizer as in the main experiments. For Arabic, we train a new tokenizer following the same recipe on a subset of our Arabic data from CulturaX.

For the Arabic downstream tasks, we evaluate on the OALL (Elfilali et al., 2024) benchmark suite using `lighteval` (Fourrier et al., 2023). The individual benchmarks are ACVA (Huang et al., 2023), AlGhafa (Almazrouei et al., 2023), and translated Arabic benchmarks from AlGhafa-T based on: MMLU (Koto et al., 2024; Hendrycks et al., 2021), EXAMS (Hardalov et al., 2020), ARC-Challenge (Clark et al., 2018b), ARC-Easy (Clark et al., 2018b), BOOLQ (Clark et al., 2019), COPA (Roemmele et al., 2011), HellaSwag (Zellers et al., 2019), OpenBookQA (Mihaylov et al., 2018), PIQA (Bisk et al., 2020), RACE (Lai et al., 2017), SciTail (Welbl et al., 2017), TOXIGEN (Hartvigsen et al., 2022). We report a macro average as in OALL (Elfilali et al., 2024) counting the benchmarks in AlGhafa-T individually. We provide further details in Appendix D.

## 4. Results & Analysis

Now we present and discuss the results of our experiments. We analyze pure vs. mixed-precision `bfloat16` in Section 4.1, tokenizer swapping in Section 4.2, and the influence of the target language in Section 4.3.

### 4.1. Analysis: Pure `bfloat16` vs. Mixed Precision

We first investigate whether pure `bfloat16` training is a viable alternative to mixed-precision training for continued pretraining for language adaptation of LLMs.

**Training efficiency gains.** First, we establish the training efficiency gains of pure `bfloat16` training over using mixed precision. Especially in settings with a tight academic compute budget, not having to store the full-precision master

copy of the model weights required for mixed-precision training can yield significantly faster training. We compare the training efficiency of pure and mixed-precision `bfloat16` in Table 2. Since our actual training runs used a mix of different hardware depending on availability, we run a comparable benchmark inspired by Hagemann et al. (2023) using Nvidia H100 80GB GPUs. In general, mixed-precision `bfloat16` training requires much more aggressive memory saving techniques to fit Mistral-7B into GPU memory[6], such as using a micro-batch size of 1, full FSDP sharding (ZeRO Stage 3), or activation checkpointing.

For a single 80GB GPU, mixed-precision `bfloat16` training was impossible even when applying all memory saving techniques due to running out-of-memory (OOM). When using only two parallel 80GB GPUs, it became necessary to use a paged AdamW implementation (Dettmers et al., 2023) or to sync gradients at every step during gradient accumulation[7]. Otherwise, mixed-precision `bfloat16` training was impossible due to OOM. In this setting using two 80GB GPUs, pure `bfloat16` was 39.2% faster than using mixed precision. In our setting with four parallel 80GB GPUs, pure `bfloat16` was 31.0% faster than using mixed precision. We note that a large part of this speedup is mainly enabled by not having to apply activation checkpointing to fit the model during training and delaying the syncing of gradients during gradient accumulation. When comparing pure `bfloat16` using the same training configuration as the most efficient mixed-precision run, pure `bfloat16` was only 11.4% faster.[8]

We also benchmark using eight parallel GPUs for reference, although this is arguably stretching a tight academic compute budget. Here, pure `bfloat16` is also 29.8% faster than mixed-precision training. Although we do not have access to sufficient compute resources to benchmark this in our setup, with sufficiently many parallel devices, mixed-precision training does not require activation checkpointing to efficiently train large-scale models, eliminating one of the main benefits of pure `bfloat16`. This highlights the benefits of pure `bfloat16` especially in settings with limited total GPU memory due to a lower number of parallel devices. In such settings, mixed-precision training is either impossible or requires computationally expensive memory-saving techniques, which pure `bfloat16` does not require to the same extent.

---

[6]By this, we refer to all states necessary during training: model weights, gradients, optimizer states, and activations

[7]Using `torch_model.no_sync()` during gradient accumulation is a common optimization to reduce communication overhead but results in larger maximum memory usage when coupled with FSDP since the gradients will not be sharded for that period.

[8]See the full benchmark results in Appendix E.

| bfloat16 | Tokenizer | NLL at % of training | | | | | | |
|---|---|---|---|---|---|---|---|---|
| | | 0% | 10% | 30% | 50% | 70 % | 90% | 100% |
| mixed | German | 5.84 | 1.96 | 1.76 | 1.67 | 1.60 | 1.56 | 1.55 |
| pure | German | 5.84 | 1.99 | 1.76 | 1.67 | 1.61 | 1.59 | 1.59 |
| mixed | original | 2.56 | 1.96 | 1.79 | 1.70 | 1.62 | 1.58 | 1.57 |
| pure | original | 2.56 | 1.98 | 1.79 | 1.69 | 1.62 | 1.60 | 1.60 |

Table 3: Word-normalized negative log-likelihood (NLL) of a held-out test set throughout continued pretraining of Mistral-7B on German text.

| bfloat16 | Layer type | Avg. parameter change |
|---|---|---|
| mixed | RMSNorm | 0.0048 |
| pure | RMSNorm | 0.000004 |
| mixed | not RMSNorm | 0.0015 |
| pure | not RMSNorm | 0.0012 |

Table 4: Average change of parameter values at the end of training compared to their starting values depending on layer type (RMSNorm or others) and pure or mixed-precision `bfloat16` training.

**Loss and downstream task results.** Comparing the loss over the course of continued pretraining of pure `bfloat16` and mixed-precision `bfloat16` training in Table 3, we see that pure and mixed-precision `bfloat16` training are very close. However, at the very beginning (after 10% of training steps) and towards the end (from 90% of training steps), we find that mixed precision achieves a slightly lower loss. This is very likely an artifact of our cosine training schedule with linear warmup: at the very beginning and end, the learning rate is very small and results in small weight updates, which are problematic in pure `bfloat16` training as we will discuss below. We also compare the downstream task performance of pure `bfloat16` and mixed-precision `bfloat16` training in Table 5. Interestingly, pure `bfloat16` outperforms mixed-precision `bfloat16` in our downstream task evaluations both in German and English. Note that pure `bfloat16` training effectively acts as a regularizer that zeroes out updates in the optimizer step that are smaller than some $\epsilon$ dependent on the magnitude of the to-be-updated weight according to the formula outlined in Section 2. We analyze this below.

**The effects of pure `bfloat16` numerics.** As shown in Figure 2, the parameter values of RMSNorm layers in Mistral-7B are particularly large, which results in a larger $\epsilon$ below which updates are squashed to zero in pure `bfloat16` training. We further investigate this in Table 4, reporting the average change in parameter values at the end of training compared to their starting value for pure and mixed-precision `bfloat16` training. We report the RMSNorm layers and other layers separately. As expected according to

| Tokenizer | bfloat16 | German translations (Plüster, 2023) | | | | German test splits | | Avg. |
|---|---|---|---|---|---|---|---|---|
| | | MMLU | HellaSwag | TruthfulQA | ARC | LAMBADA | PAWS-X | |
| | | Main experiments (see Section 3.1) | | | | | | |
| German | mixed | 33.6 | **60.0** | 37.5 | 43.0 | 40.3 | 62.3 | 46.1 |
| German | pure | **35.9** | 59.7 | **39.4** | **44.1** | **40.6** | **63.6** | **47.2** |
| original | mixed | 32.6 | **59.5** | **43.0** | 40.8 | 38.8 | 63.5 | 46.4 |
| original | pure | **37.2** | 59.4 | 39.2 | **41.6** | **39.3** | **63.9** | **46.8** |
| | | Improved hindsight runs (see Section 3.2) | | | | | | |
| German | pure | 43.5 | 63.4 | **39.8** | **47.9** | **37.9** | **62.5** | **49.1** |
| German | pure++[†] | **43.6** | **63.5** | 39.5 | 46.7 | **37.9** | **62.5** | 48.9 |

Table 5: Effectiveness of models based on Mistral-7B on German downstream tasks. The best result in each section is **bolded** and the overall best result of the main experiments is additionally **underlined**. [†]: for pure++ bfloat16, mixed precision was used just for the final annealing phase of the learning rate schedule.

bfloat16 numerics, the RMSNorm layers receive almost no updates in pure bfloat16 training due to their larger weight values. We see that pure bfloat16 also reduces the average parameter change for other layers, but this effect is much less pronounced.

It is not clear that the regularizing effect of pure bfloat16 training is a desired behavior, even though it results in better downstream task performance in our experiments. However, given the large training efficiency boost possible through pure bfloat16 training when on a tight academic compute budget, we believe that this potentially undesired side-effect is a worthwhile tradeoff to consider. We stress that we perform a data-matched comparison between pure and mixed-precision bfloat16. A compute-matched comparison, where pure bfloat16 will train for significantly more steps, would be even more favorable for pure bfloat16 training. Additionally, we can enable mixed precision or float32 just for the parameters with large values (in the case of Mistral-7B these are the RMSNorms) to offset the majority of bfloat16 regularization impact.

In terms of loss, we do observe slightly better performance of mixed precision at the very end and beginning of training due to the linear warmup and cosine decay of the learning rate. As the final annealing phase has been shown to be important for model performance (Ibrahim et al., 2024), we further study this in our hindsight runs, which are trained in pure bfloat16. We employ an "infinite" learning rate schedule (Ibrahim et al., 2024; Zhai et al., 2022) with a prolonged constant learning rate before the final annealing down to a smaller value. We use the same checkpoint before the beginning of the annealing phase and continue training with either pure or mixed-precision bfloat16. Using mixed precision during the annealing phase is denoted as pure++ in Table 5 and Table 7. Interestingly, we do not see any

conclusive advantage of using mixed precision during the annealing phase. We conclude that fully pure bfloat16 continued training is indeed viable even during the low learning rate annealing phases of a learning rate schedule.

### 4.2. Analysis: Tokenizer Swapping

The next research question we investigate in this work are the benefits of tokenizer swapping for language adaption of pretrained LLMs. This question deserves special attention in the tight academic budget setting, as there is less compute spent on re-learning the new embeddings.

**On comparing loss between different tokenizers.** Comparing the loss (negative log-likelihood) across different tokenizers is not straightforward. Different tokenizers produce different numbers of tokens for the same text. The conventionally reported loss is the summed negative log-likelihood normalized by the number of tokens – a direct advantage for worse tokenizers that produce more tokens for the same text.

Instead, we choose to normalize by a tokenizer-independent quantity: the number of (white-space split) words. This choice is arbitrary in some respects as it is just a constant for all methods. Instead, we could also normalize by the number of characters or bytes (see Gao, 2021; Forsythe, 2023; Gao et al., 2020). Choosing words as the normalizing factor has the advantage of yielding a similar value range to the familiar token normalization. The word-normalized negative log-likelihood is computed over the same chunks of text for each model. We report the token-normalized log-likelihood in Appendix C.

**Embedding re-initialization.** When swapping tokenizers for language adaptation, it has been shown that a good re-

initialization of the new embedding matrix is crucial (Minix-hofer et al., 2022; Ostendorff & Rehm, 2023; Dobler & de Melo, 2023; Downey et al., 2023). We evaluate various embedding initialization methods and find that FO-CUS (Dobler & de Melo, 2023) performs best for German. For Arabic in our hindsight study, all methods showed sub-optimal performance, so we use FOCUS coupled with an additional short gradient descent training phase of the embeddings with the rest of the model frozen for 100 steps (roughly 1.5% of total training steps). Full details are provided in Appendix B.

**Results of tokenizer swapping.** In Table 3, we report the word-normalized negative log-likelihood of a held-out test set throughout training. We see that directly after re-initializing the new embeddings, tokenizer swapping performs worse than keeping the original tokenizer. This is expected, as no gradient descent training has yet been performed on the newly initialized embedding matrix. Tokenizer swapping quickly catches up at the next evaluation interval and obtains a slight advantage during further training. The initial gap could be improved by a short and computationally cheap initial training phase during which only the new embeddings are trained (de Vries & Nissim, 2021).

We report downstream task results on a suite of German benchmarks in Table 5. We do not see a clear trend of better performance with or without tokenizer swapping. Note that for inference on downstream tasks in the target language, however, tokenizer swapping will be computationally more efficient, as the new tokenizer will produce fewer tokens for the same text (Yamaguchi et al., 2024).[9] In Table 6, we also report the average performance on English downstream tasks. As expected, we find that replacing the original tokenizer with an exclusively German tokenizer leads to diminished results on English. Thus, whenever performance in the source language is important, the new tokenizer should perhaps retain a larger share of the original tokens.

The base model's tokenizer already contains some German tokens[10], as well as many English tokens also used in German. Our experiments hence provide a rough lower bound on the effectiveness of tokenizer swapping. We expect much larger benefits for unseen and low-resource languages. Even though downstream tasks on German do not improve significantly with a specialized tokenizer, our results demonstrate the viability of re-learning an embedding matrix after tokenizer swapping even when on a tight academic compute budget. In contrast, Zhao et al. (2024b) find that vocabulary extension (instead of full tokenizer swapping) actually un-

---

[9]This can lead to significant computational savings: In our evaluation on Arabic, Mistral-7B took over 5 times as long as a model with our custom Arabic tokenizer due to poorer tokenizer fertility.

[10]Such as _Jahrhunderts or meisterschaft.

| Compute Budget | Tokenizer | German Avg. | English Avg. |
|---|---|---|---|
| small (ours) + mixed bfloat16 | German | 46.1 | 44.6 |
| small (ours) + pure bfloat16 | German | **47.2** | 44.9 |
| small (ours) + mixed bfloat16 | original | 46.4 | 46.6 |
| small (ours) + pure bfloat16 | original | 46.8 | **47.1** |
| 8x larger (LeoLM) | original | **51.8** | 56.9 |
| no further training (Mistral-7B) | original | 51.2 | **62.4** |

Table 6: Effectiveness of models based on Mistral-7B on benchmark suites in English and German. The English benchmarks used are MMLU, HellaSwag, TruthfulQA, and ARC.

| | ACVA | AlGhafa | MMLU-AR | AlGhafa-T | Macro Avg. |
|---|---|---|---|---|---|
| Arabic Mistral-7B (w/ tokenizer swapping & pure bfloat16, see Section 3.2) | | | | | |
| pure | **73.2** | 62.5 | **37.9** | 52.4 | 53.6 |
| pure++[†] | 70.7 | **63.3** | 37.7 | **52.8** | **53.8** |
| | | Baselines | | | |
| Mistral-7B | 63.0 | **57.7** | **33.9** | **46.3** | **47.4** |
| AceGPT-7B | 71.0 | 52.6 | 27.0 | 44.6 | 45.8 |
| Llama 2-7B | 66.3 | 45.6 | 27.4 | 41.3 | 42.4 |

Table 7: Results on Arabic downstream task suites. The average is a macro-average that includes the individual benchmarks in AlGhafa-T. [†]: For pure++ bfloat16, mixed precision was used just for the final annealing phase of the learning rate schedule.

derperforms compared to keeping the original vocabulary for tight compute budget continued pretraining on Chinese. However, we note that our experiments for tokenizer swapping are compute-matched but not data-matched: since the new tokenizer encodes the training data more efficiently, more total text was seen during continued pretraining for the same number of tokens. Our experiments do not suggest that this increased efficiency translates to better performance.

### 4.3. Analysis: Target Language

From the results in Table 6, we have to conclude that the resulting models from our main experiments do not show good downstream task performance. Strikingly, the base Mistral-7B model achieves better downstream task results than our adapted German variants, even on German benchmarks. Mistral-7B has already seen German during pretraining and its tokenizer contains German tokens, which allows us to roughly lower bound language adaptation effectiveness, but for most other languages it would be a pessimistic estimate. Therefore, we further adapt Mistral-7B to Arabic alongside German in an additional hindsight study. Also, the models trained in our hindsight study were trained using higher quality data from CulturaX rather than OSCAR23.01 and used several other improved training techniques (see Section 3.2).

**German downstream results.** The German hindsight study model using the custom German tokenizer has an average improved downstream task performance of two percentage points over the best model from the main experiments using the exact same computational budget. This demonstrates the importance of data quality and our training improvements such as preventing cross-document attention contamination via an adjusted attention mask. However, the model still performs worse than the base Mistral-7B and its adapted German LeoLM version on German downstream tasks.

**Arabic downstream results.** In contrast, the Arabic hindsight study models perform significantly better than the base Mistral-7B model on Arabic benchmarks (reported in Table 7). Interestingly, even though the goal of our study is analysis rather than achieving state-of-the-art, our Arabic models also outperforms AceGPT-7B (Huang et al., 2023), which is a continued pretraining of Llama 2 on Arabic for $3.75\times$ more tokens than our models but without tokenizer swapping. Note that the training recipes for our German and Arabic hindsight study models were exactly the same – highlighting that language adaptation is especially useful for underrepresented languages.

**When is language adaptation on a tight academic compute budget viable?** Although the training data of Mistral-7B is not open, due to its multilingual performance and the existence of German tokens in its tokenizer, we can conclude that it was pretrained multilingually and that German had a significant share. Intuitively, specializing such a model on a single language might boost performance, e.g., by removing *curse of multilinguality* (Conneau et al., 2020) capacity bottlenecks. Our experiments with German as a target language show that continued pretraining to focus model capacity just on the target language is not always beneficial. This is likely because German was already well-represented. Recent language adaptation attempts for lesser-resourced or less well-represented languages than German, e.g., Polish (Ruciński, 2024) or Chinese (Zhao et al., 2024b), do report improvements in the target language compared to the base model with similar computational budgets. LeoLM used eight times as much compute as our study and obtains a minor increase in German task performance of 0.6 percentage points.

Our results on Arabic highlight that target languages that were not dominant or unseen during pretraining can indeed benefit from language adaptation on a tight academic compute budget. In our setup, we focus only on monolingual performance in the target language and do not include English or code in the training data. Recent work has shown that the inclusion of such data can boost performance even in the target language (Muennighoff et al., 2023; Csaki et al.,

2023). However, it is not clear if such benefits will also materialize on a tight academic compute budget.

## 5. Related Work

**Language adaptation of pretrained LLMs.** Adapting strong pretrained LLMs such as Llama (Touvron et al., 2023a;b) or Mistral-7B (Jiang et al., 2023b) to other languages has recently had a surge in popularity (e.g., Plüster et al., 2023; Pires et al., 2023; Cui et al., 2023; Nguyen et al., 2023; Alves et al., 2024). However, these studies use significantly more compute than our setting and many either do not specialize the tokenizer or do not use strong embedding initialization methods. Csaki et al. (2024) adapt pretrained LLMs to a variety of languages and find that specializing tokenizers through vocabulary expansion improves fertility but not downstream task results. Recent work has shown that mixing in English during continued pretraining helps downstream task performance in the target languages (Csaki et al., 2023; Zhao et al., 2024b), whereas Ibrahim et al. (2024) find that skewing the data mix towards the target language helps adaptation at the cost of more forgetting. Muennighoff et al. (2023) show in data-constrained experiments on English that individual samples can be repeated up to four times and mixing in code data can improve downstream task performance.

**Efficient low-precision training.** The necessity of `bfloat16` mixed precision requiring the storage of `float32` optimizer states and a master copy of the model weights (Micikevicius et al., 2017; Kalamkar et al., 2019) has been studied in the literature. Rae et al. (2021) observed degraded performance and stale layers for pure `bfloat16` training of Transformer language models from scratch but achieved better results when applying stochastic rounding to the weight updates and casting optimizer states to `float32`. Lewandowski & Kosson (2023) propose a variant of mixed precision `bfloat16` where – instead of keeping a `float32` master copy of the weights – only a difference to a `float16` representation is stored alongside an optimization procedure optimized for memory pressure. Motivated by LoRA (Hu et al., 2021), Zhao et al. (2024a) perform the weight update in a low-rank approximation space, significantly reducing memory cost. Zamirai et al. (2020) train Transformer models from scratch in pure `bfloat16` and find that stochastic rounding of the weight updates was necessary to recover the performance of using `float32` optimizer states and master copy of the weights. Despite this, the most popular choice for training language models in many popular libraries remains mixed-precision `bfloat16` (Lewandowski & Kosson, 2023). We show that for continued pretraining, pure `bfloat16` training is viable without any further modifications such as stochastic rounding.

## 6. Conclusion

Based on our results, we draw the following final conclusions.

**Pure `bfloat16`.** Training in pure `bfloat16` (rather than using mixed precision) is viable for continued pretraining and enables significantly faster training when only a few parallel devices are available. Pure `bfloat16` training does come with certain caveats: We find that for Mistral-7B, the RMSNorm weights (which are larger than most other weights) are not updated in pure `bfloat16` training due to the numerics of `bfloat16`. However, this has no clear drawbacks in terms of loss and downstream task performance.

**Tokenizer swapping.** Tokenizer swapping coupled with a good embedding initialization is viable and performs at least on par with keeping the original vocabulary but did not significantly improve downstream task results for German. We note that Mistral-7B's tokenizer already contains some German tokens. For Arabic, which was an unseen script in the original tokenizer, our adapted model with tokenizer swapping fares significantly better than the base model without language adaptation. Our results demonstrate that re-learning the embeddings for a new tokenizer is viable even when on a tight compute budget.

**Tight compute budgets for language adaptation.** Adapting Mistral-7B to German on a tight compute budget performed worse than the base Mistral-7B for German downstream tasks, irrespective of tokenizer swapping and training precision. Adapting to Arabic however significantly increased Arabic downstream task performance. This highlights that language adaptation is not always beneficial but can be, especially if the target language was not well-represented. We believe that our findings regarding pure `bfloat16` training and tokenizer swapping can help inform future language adaptation efforts towards more efficiency.

## Limitations

We limit our study to a single LLM (Mistral-7B). This is motivated by both the checkpoint's popularity and the popularity of its parameter-size class (seven billion parameters). However, it is not certain that model-specific findings, such as RMSNorm updates being flushed to zero in pure `bfloat16` training, will transfer to other models. Our general findings on weight updates in pure `bfloat16` to large values are expected to generalize. Also, while we add Arabic as a second language in our hindsight study, we consider only German and Arabic in our experiments. Although less significant for our findings on pure `bfloat16`, the efficacy of tokenizer swapping will likely vary depending on the target language.

Additionally, a large part of the training efficiency gains of pure `bfloat16` training compared to mixed-precision training materialize mainly when the training is constrained by GPU memory. For models smaller than Mistral-7B that might not require sharded training, these benefits will be less pronounced.

We note that we analyze the viability of pure `bfloat16` only for continued pretraining, where the weights effectively start with an initialization (reasonably) close to their optimized value. We cannot draw the same conclusion for pretraining from scratch, as the training dynamics in the initial phase of pretraining from randomly initialized weights might have a much larger dependence on using high-precision optimizer states and a master copy of the model weights to reduce numerical errors.

We consider only full-parameter training and do not evaluate the use of techniques such as LoRA (Hu et al., 2021) due to the low-rank limitation they impose on the weight update. Whether LoRA(-like) methods are sufficient for language adaptation should be evaluated seperately.

## Acknowledgements

The authors acknowledge support by the German Federal Ministry for Education and Research (BMBF) through the project «KI-Servicezentrum Berlin Brandenburg» (01IS22092). We also thank the reviewers of the WANT@ICML workshop for their helpful comments.

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

| Embedding initialization method | Loss on test set |
|---|---|
| FOCUS (Dobler & de Melo, 2023) | **5.8** |
| WECHSEL (Minixhofer et al., 2022) | 8.7 |
| Heuristics (Downey et al., 2023) | 7.3 |
| $\mathcal{N}(\text{orig\_embs\_mean}, \text{orig\_embs\_std})$ | 11.9 |
| Random-Assign | 19.3 |
| $\mathcal{N}(0, 0.02)$ | 26.4 |

Table 8: Effectiveness of different embedding initialization methods *directly after initialization* (no training performed), measured by the cross-entropy loss on a test set. Since we are comparing across the same tokenizer, word-normalization is not necessary.

## A. Further Training Details

We publish our training code at https://github.com/konstantinjdobler/tight-budget-llm-adaptation. We use PyTorch FSDP for sharded training, and we rely on the `lightning` package for the implementation of distributed and mixed-precision training. We employ document packing and concatenate samples separated by an [EOS] token until the block size is filled. In our hindsight runs, we instead prepend a [BOS] token and adjust the attention mask so that cross-document attention is prevented. We enable TF32 computation for `float32` via `torch.set_float32_matmul_precision("high")`.

For pure `bfloat16` training, the Mistral-7B implementation from HuggingFace implements some small parts of the forward pass in `float32`, which we do not modify. Specifically, these are the input variance calculations in the RMSNorm layers and the final softmax in the language modeling head. All weights and optimizer states are still in `bfloat16` for pure `bfloat16`.

For the "infinite" learning rate schedules in our hindsight study, we follow Ibrahim et al. (2024) and use a 1% warmup up to a maximum learning rate of $3 \times 10^{-5}$, then a cosine decay for 60% of steps to $1.65 \times 10^{-5}$, followed by a constant learning rate for 25% of steps, and finally annealing down to $2 \times 10^{-6}$ for the last 14% of steps.

We use Docker images to conduct our training in a fully-reproducible environment, which are published alongside lockfiles with the pinned package versions that we used.

## B. Embedding Re-initialization

When swapping tokenizers for language adaptation, it has been shown that harnessing the compute spent on the original embedding matrix via a good re-initialization of the new embedding matrix is crucial (Minixhofer et al., 2022; Ostendorff & Rehm, 2023; Dobler & de Melo, 2023; Downey et al., 2023). We evaluate FOCUS (Dobler

& de Melo, 2023) and WECHSEL (Minixhofer et al., 2022), two initialization methods based on mappings to the old embedding space. Furthermore, we evaluate a heuristics-based approach (Downey et al., 2023), initializing from $\mathcal{N}(\text{original\_embs\_mean}, \text{original\_embs\_std})$ or $\mathcal{N}(0, 0.02)$, and simply copying the old embeddings, effectively randomly assigning old embeddings to new tokens.[11]

We evaluate all the mentioned methods directly after initialization without any further training and report the results in Table 8. We find that out of these methods, FOCUS provided the best results directly after initialization. Therefore, we utilize FOCUS for all experiments with tokenizer swapping. Note that these results might differ for different target languages and the gap between these methods is reduced by continued pretraining.

For Arabic in our hindsight study, all methods showed suboptimal performance, so we use FOCUS coupled with an additional short gradient descent training phase of the embeddings with the rest of the model frozen for 100 steps (roughly 1.5% of total training steps). Specifically, FOCUS likely underperformed because we deliberately did not add English tokens to the new tokenizer, leading to a very small – and "low-quality" overlap with the original vocabulary, which FOCUS relies on. This could be ameliorated by including more tokens from the original vocabulary or employing a bilingual dictionary.

Very recently – after our experiments had finished – Minixhofer et al. (2024) proposed ZeTT, which utilizes a trained hypernetwork to predict embeddings for a new tokenizer. We briefly discuss this method since it yielded very promising initial results. We found that ZeTT achieves a lower initial loss than FOCUS for German and Arabic. For Arabic specifically, it is the only method that performed significantly better than random. ZeTT requires a hypernetwork trained specifically for the base model, which is not computationally sensible as a one-time effort for the embedding initialization. However, the authors do provide such a hypernetwork for Mistral-7B and select other popular base models. The computational cost of predicting the new embeddings using the hypernetwork is negligible. If such a hypernetwork is available, using ZeTT could further improve the performance of tokenizer swapping over our current results using FOCUS.

## C. Token-normalized Loss

Alongside the word-normalized negative log-likelihood reported in Table 3 in the main body of the paper, we report the token-normalized negative log-likelihood (the conventional cross-entropy loss) in Figure 3. Note that this is not a

---

[11]Since our new tokenizer has a larger vocabulary than the original one, we initialize the remaining tokens from $\mathcal{N}(0, 0.02)$.

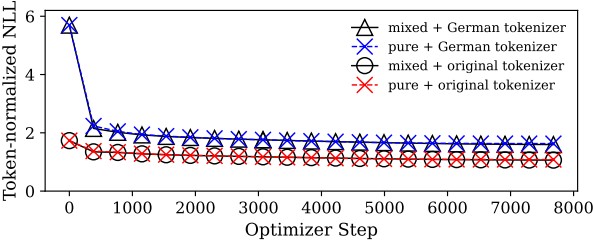

Figure 3: Token-normalized negative log-likelihood (conventional cross-entropy loss) of a held-out test set throughout continued pretraining of Mistral-7B on German text. We compare pure and mixed-precision `bfloat16` training. Additionally, we compare swapping the original tokenizer of Mistral-7B with a specialized German tokenizer.

fair comparison and we only provide this for completeness, as Mistral-7B's original tokenizer will produce more tokens for the same text, which directly results in a lower loss as the loss is divided by the number of tokens.

## D. Full Downstream Task Results

We report the full downstream task results for all models on German (in Table 9) and on English (in Table 10).

For German benchmark evaluation, we use the benchmark suite provided by Plüster (2023), which was also used for LeoLM. We use their fork of the `lm-eval-harness` (Gao et al., 2023) to evaluate our checkpoints. The individual datasets used are translated versions of MMLU (Hendrycks et al., 2021), TruthfulQA (Lin et al., 2021), ARC (Clark et al., 2018a), HellaSwag (Zellers et al., 2019), as well as PAWS-X (Yang et al., 2019) and LAMBADA-OpenAI (Paperno et al., 2016; Radford et al., 2019).

For the Arabic benchmarks based on OALL (Elfilali et al., 2024), we used ACVA (Huang et al., 2023), AlGhafa (Al-mazrouei et al., 2023), and the following translated benchmarks from AlGhafa-T: MMLU (Koto et al., 2024; Hendrycks et al., 2021), EXAMS (Hardalov et al., 2020), ARC-Challenge (Clark et al., 2018b), ARC-Easy (Clark et al., 2018b), BOOLQ (Clark et al., 2019), COPA (Roemmele et al., 2011), HellaSwag (Zellers et al., 2019), OpenBookQA (Mihaylov et al., 2018), PIQA (Bisk et al., 2020), RACE (Lai et al., 2017), SciTail (Welbl et al., 2017), TOX-IGEN (Hartvigsen et al., 2022). We report the full results of Arabic models for all benchmarks in Table 11. The Arabic benchmarks were evaluated using `lighteval` (Fourrier et al., 2023) in a 5-shot setting.

## E. Performance Benchmarking

We ran a performance benchmark inspired by Hagemann et al. (2023) to determine the best possible combination of micro-batch size, activation checkpointing, FSDP ZeRO sharding, syncing during gradient accumulation (not syncing is faster but incurs a memory overhead), and utilizing a paged variant of AdamW for each combination of number of GPUs and precision type. We evaluate the full Cartesian product of micro-batch sizes {1,2,4,8}, activation checkpointing {yes, no}, FSDP ZeRO sharding {full, grad_op}, gradient accumulation syncing {sync, no_sync}, and AdamW implementation {paged, no_paged}. For training on a single GPU, mixed-precision training is impossible even with all memory saving techniques maximized. For pure `bfloat16` on a single GPU, we instead manually search for the best configuration. We do not consider FSDP CPU offloading, as this did not yield acceptable compute efficiency. We report the full results of this benchmark in Table 12. For the purposes of this paper, we actually run the full benchmark to compare pure and mixed-precision `bfloat16` using the same configuration, even if a better configuration is also available for pure `bfloat16`. We do make one small optimization and exclude trials where a more memory-friendly configuration previously went OOM. To save compute, one could additionally prune benchmark trials through educated guesses of suboptimal configurations (e.g., when using eight devices, we do not need to benchmark heavy memory optimizations) or employ a binary instead of linear search. For all benchmarked configurations, we run 11 steps and exclude the first one from the reported average since it is disproportionately affected by initial one-time costs. Due to the way we extract step timings from our original training script, timings are rounded to the nearest second for step times over a minute instead of considering two decimal places.

| | MMLU | HellaSwag | TruthfulQA | ARC | LAMBADA | PAWS-X | Avg. |
|---|---|---|---|---|---|---|---|
| German tokenizer + mixed-precision `bfloat16` | 33.6 | 60.0 | 37.5 | 43.0 | 40.3 | 62.3 | 46.1 |
| German tokenizer + pure `bfloat16` | 35.9 | 59.7 | 39.4 | 44.1 | 40.6 | 63.6 | 47.2 |
| original tokenizer + mixed-precision `bfloat16` | 32.6 | 59.5 | 43.0 | 40.8 | 38.8 | 63.5 | 46.4 |
| original tokenizer + pure `bfloat16` | 37.2 | 59.4 | 39.2 | 41.6 | 39.3 | **63.9** | 46.8 |
| Mistral-7B (original) | **52.2** | 58.7 | **48.5** | 47.2 | 40.1 | 60.3 | 51.2 |
| LeoLM | 48.0 | **66.3** | 40.8 | **48.5** | **43.6** | 63.7 | **51.8** |

Table 9: Results on German downstream tasks.

| | MMLU | HellaSwag | TruthfulQA | ARC | Avg. |
|---|---|---|---|---|---|
| German tokenizer + mixed-precision `bfloat16` | 33.3 | 65.5 | 39.3 | 40.4 | 44.6 |
| German tokenizer + pure `bfloat16` | 32.8 | 66.1 | 40.4 | 40.4 | 44.9 |
| original tokenizer + mixed-precision `bfloat16` | 34.5 | 67.6 | **43.8** | 40.6 | 46.6 |
| original tokenizer + pure `bfloat16` | 36.8 | 68.7 | 39.5 | 43.4 | 47.1 |
| Mistral-7B (original) | **63.5** | **83.3** | 42.6 | **60.3** | **62.4** |
| LeoLM | 55.1 | 77.8 | 42.9 | 51.9 | 56.9 |

Table 10: Results on English downstream tasks.

| | ACVA | AlGhafa | MMLU-AR | openbook-qa-ext-ar | race-ar | arabic-exams | hellaswag-okapi-ar | piqa-ar | arc-easy-ar | boolq-ar | arc-challenge-okapi-ar | xstory-cloze | copa-ext-ar | sciq-ar | toxigen-ar | Avg. |
|---|---|---|---|---|---|---|---|---|---|---|---|---|---|---|---|---|
| *Arabic Mistral-7B (w/ tokenizer swapping & pure `bfloat16`, see Section 3.2)* | | | | | | | | | | | | | | | | |
| Arabic pure | **73.2** | 62.5 | **37.9** | 53.7 | 44.2 | 36.7 | 26.0 | 57.8 | **62.7** | 74.3 | **49.7** | 66.2 | 62.2 | 66.7 | 42.2 | 54.4 |
| Arabic pure++ | 70.7 | **63.3** | 37.7 | **54.3** | **44.3** | **37.1** | 25.9 | **58.5** | 62.3 | **75.4** | 49.5 | **66.6** | **64.4** | 56.7 | 52.5 | **54.6** |
| *Baselines* | | | | | | | | | | | | | | | | |
| Mistral-7B | 63.0 | 57.7 | 33.9 | 41.4 | 39.1 | 31.7 | 25.7 | 57.0 | 44.8 | 64.7 | 36.3 | 52.3 | 57.8 | **72.4** | 38.7 | 47.8 |
| AceGPT-7B | 71.0 | 52.6 | 27.0 | 39.8 | 30.7 | 25.9 | **26.3** | 53.0 | 47.0 | 65.4 | 35.8 | 62.9 | 60.0 | 64.7 | 41.7 | 46.9 |
| Llama 2-7B | 66.3 | 45.6 | 27.4 | 33.3 | 28.9 | 24.8 | 25.0 | 51.3 | 27.8 | 65.3 | 25.9 | 50.2 | 56.7 | 58.7 | **56.7** | 42.9 |

Table 11: Results on Arabic downstream tasks.

Table 12: Performance benchmarking results grouped by (precision, # GPUs), sorted by Total H100 80GB GPU Hours. We abbreviate `mb := micro-batch size`, `ckpt := activation checkpointing`, and `sharding := FSDP ZeRO stage sharding`. FSDP `grad_op` shards only gradients and optimizer states, whereas `full` additionally shards the model weights.

| Precision | # GPUs | (mb, ckpt, sharding) | Max. CUDA RAM | Step time | Est. total GPU Hours |
|---|---|---|---|---|---|
| mixed-precision bfloat16 | 1 | OOM | OOM | N/A | N/A |
| pure bfloat16 | 1 | (1, no, N/A, N/A, paged) | 55.85 GB | 107s | 228.27 hours |
| mixed-precision bfloat16 | 2 | (4, yes, full, sync, paged) | 48.96 GB | 74.30s ± 0.90s | 317.01 hours |
| mixed-precision bfloat16 | 2 | (4, yes, full, no_sync, paged) | 63.46 GB | 75.60s ± 1.50s | 322.56 hours |
| mixed-precision bfloat16 | 2 | (4, yes, grad_op, sync, paged) | 75.13 GB | 77.60s ± 0.92s | 331.09 hours |
| mixed-precision bfloat16 | 2 | (2, yes, grad_op, sync, paged) | 67.41 GB | 81.70s ± 0.46s | 348.59 hours |
| mixed-precision bfloat16 | 2 | (2, yes, full, sync, no_paged) | 74.16 GB | 81.90s ± 0.30s | 349.44 hours |
| mixed-precision bfloat16 | 2 | (2, yes, full, sync, paged) | 41.23 GB | 83.60s ± 1.80s | 356.69 hours |
| mixed-precision bfloat16 | 2 | (2, yes, full, no_sync, paged) | 55.73 GB | 84.00s ± 0.00s | 358.40 hours |
| mixed-precision bfloat16 | 2 | (1, yes, full, sync, paged) | 51.88 GB | 94.10s ± 0.30s | 401.49 hours |
| mixed-precision bfloat16 | 2 | (1, yes, grad_op, sync, paged) | 63.55 GB | 96.10s ± 0.30s | 410.03 hours |
| mixed-precision bfloat16 | 2 | (1, yes, full, sync, no_paged) | 73.62 GB | 97.40s ± 0.49s | 415.57 hours |
| mixed-precision bfloat16 | 2 | (1, yes, full, sync, paged) | 37.38 GB | 98.90s ± 0.70s | 421.97 hours |
| mixed-precision bfloat16 | 2 | (1, no, full, sync, paged) | 78.11 GB | 120.90s ± 5.63s | 515.84 hours |
| mixed-precision bfloat16 | 2 | (8, yes, grad_op, no_sync, no_paged) | OOM | N/A | N/A |
| mixed-precision bfloat16 | 2 | (4, no, grad_op, sync, paged) | OOM | N/A | N/A |
| mixed-precision bfloat16 | 2 | (8, no, grad_op, no_sync, no_paged) | OOM | N/A | N/A |
| mixed-precision bfloat16 | 2 | (4, no, grad_op, no_sync, paged) | OOM | N/A | N/A |
| mixed-precision bfloat16 | 2 | (8, no, full, no_sync, paged) | OOM | N/A | N/A |
| mixed-precision bfloat16 | 2 | (2, yes, full, no_sync, no_paged) | OOM | N/A | N/A |
| mixed-precision bfloat16 | 2 | (1, no, full, no_sync, no_paged) | OOM | N/A | N/A |
| mixed-precision bfloat16 | 2 | (8, yes, full, sync, paged) | OOM | N/A | N/A |
| mixed-precision bfloat16 | 2 | (2, no, grad_op, sync, no_paged) | OOM | N/A | N/A |
| mixed-precision bfloat16 | 2 | (2, no, grad_op, no_sync, paged) | OOM | N/A | N/A |
| mixed-precision bfloat16 | 2 | (1, yes, grad_op, no_sync, no_paged) | OOM | N/A | N/A |
| mixed-precision bfloat16 | 2 | (8, yes, grad_op, no_sync, paged) | OOM | N/A | N/A |
| mixed-precision bfloat16 | 2 | (8, no, grad_op, sync, paged) | OOM | N/A | N/A |
| mixed-precision bfloat16 | 2 | (2, no, grad_op, no_sync, no_paged) | OOM | N/A | N/A |
| mixed-precision bfloat16 | 2 | (8, no, full, no_sync, no_paged) | OOM | N/A | N/A |
| mixed-precision bfloat16 | 2 | (1, no, grad_op, no_sync, paged) | OOM | N/A | N/A |
| mixed-precision bfloat16 | 2 | (2, yes, grad_op, no_sync, paged) | OOM | N/A | N/A |
| mixed-precision bfloat16 | 2 | (8, yes, full, no_sync, paged) | OOM | N/A | N/A |
| mixed-precision bfloat16 | 2 | (4, no, full, no_sync, no_paged) | OOM | N/A | N/A |
| mixed-precision bfloat16 | 2 | (4, yes, full, no_sync, no_paged) | OOM | N/A | N/A |
| mixed-precision bfloat16 | 2 | (8, yes, grad_op, sync, paged) | OOM | N/A | N/A |
| mixed-precision bfloat16 | 2 | (8, yes, grad_op, sync, no_paged) | OOM | N/A | N/A |
| mixed-precision bfloat16 | 2 | (8, no, grad_op, no_sync, paged) | OOM | N/A | N/A |
| mixed-precision bfloat16 | 2 | (2, no, full, sync, paged) | OOM | N/A | N/A |
| mixed-precision bfloat16 | 2 | (1, no, grad_op, sync, paged) | OOM | N/A | N/A |
| mixed-precision bfloat16 | 2 | (4, yes, grad_op, sync, no_paged) | OOM | N/A | N/A |
| mixed-precision bfloat16 | 2 | (1, no, grad_op, sync, no_paged) | OOM | N/A | N/A |
| mixed-precision bfloat16 | 2 | (8, yes, full, no_sync, no_paged) | OOM | N/A | N/A |
| mixed-precision bfloat16 | 2 | (1, yes, full, no_sync, no_paged) | OOM | N/A | N/A |
| mixed-precision bfloat16 | 2 | (2, no, grad_op, sync, paged) | OOM | N/A | N/A |
| mixed-precision bfloat16 | 2 | (8, yes, full, sync, no_paged) | OOM | N/A | N/A |
| mixed-precision bfloat16 | 2 | (4, no, full, sync, paged) | OOM | N/A | N/A |
| mixed-precision bfloat16 | 2 | (1, no, full, sync, no_paged) | OOM | N/A | N/A |
| mixed-precision bfloat16 | 2 | (4, no, full, no_sync, paged) | OOM | N/A | N/A |
| mixed-precision bfloat16 | 2 | (1, yes, grad_op, sync, no_paged) | OOM | N/A | N/A |
| mixed-precision bfloat16 | 2 | (2, yes, grad_op, no_sync, no_paged) | OOM | N/A | N/A |
| mixed-precision bfloat16 | 2 | (4, yes, grad_op, no_sync, no_paged) | OOM | N/A | N/A |
| mixed-precision bfloat16 | 2 | (4, yes, full, sync, no_paged) | OOM | N/A | N/A |
| mixed-precision bfloat16 | 2 | (2, no, full, no_sync, paged) | OOM | N/A | N/A |
| mixed-precision bfloat16 | 2 | (2, no, full, sync, no_paged) | OOM | N/A | N/A |
| mixed-precision bfloat16 | 2 | (8, no, grad_op, sync, no_paged) | OOM | N/A | N/A |
| mixed-precision bfloat16 | 2 | (1, no, grad_op, no_sync, no_paged) | OOM | N/A | N/A |
| mixed-precision bfloat16 | 2 | (4, no, grad_op, sync, no_paged) | OOM | N/A | N/A |
| mixed-precision bfloat16 | 2 | (1, yes, grad_op, no_sync, paged) | OOM | N/A | N/A |
| mixed-precision bfloat16 | 2 | (8, no, full, sync, paged) | OOM | N/A | N/A |
| mixed-precision bfloat16 | 2 | (4, no, grad_op, no_sync, no_paged) | OOM | N/A | N/A |
| mixed-precision bfloat16 | 2 | (1, no, full, no_sync, paged) | OOM | N/A | N/A |
| mixed-precision bfloat16 | 2 | (2, yes, grad_op, sync, no_paged) | OOM | N/A | N/A |
| mixed-precision bfloat16 | 2 | (4, yes, grad_op, no_sync, paged) | OOM | N/A | N/A |
| mixed-precision bfloat16 | 2 | (4, no, full, sync, no_paged) | OOM | N/A | N/A |
| mixed-precision bfloat16 | 2 | (2, no, full, no_sync, no_paged) | OOM | N/A | N/A |
| mixed-precision bfloat16 | 2 | (8, no, full, sync, no_paged) | OOM | N/A | N/A |
| pure bfloat16 | 2 | (1, no, grad_op, no_sync, no_paged) | 77.61 GB | 53.36s ± 0.13s | 227.67 hours |
| pure bfloat16 | 2 | (1, no, grad_op, no_sync, paged) | 63.12 GB | 54.26s ± 0.45s | 231.53 hours |
| pure bfloat16 | 2 | (2, no, full, sync, paged) | 68.59 GB | 54.91s ± 0.25s | 234.29 hours |
| pure bfloat16 | 2 | (1, no, full, no_sync, no_paged) | 64.31 GB | 57.84s ± 0.23s | 246.80 hours |
| pure bfloat16 | 2 | (1, no, grad_op, sync, no_paged) | 70.36 GB | 58.18s ± 0.22s | 248.25 hours |

| Precision | # GPUs | (mb, ckpt, sharding) | Max. CUDA RAM | Step time | Total GPU Hours |
|---|---|---|---|---|---|
| pure bfloat16 | 2 | (1, no, full, no_sync, paged) | 49.81 GB | 59.56s ± 0.84s | 254.14 hours |
| pure bfloat16 | 2 | (1, no, full, sync, paged) | 42.56 GB | 59.58s ± 0.25s | 254.23 hours |
| pure bfloat16 | 2 | (1, no, full, sync, no_paged) | 57.06 GB | 59.62s ± 0.16s | 254.37 hours |
| pure bfloat16 | 2 | (8, yes, grad_op, no_sync, no_paged) | 72.7 GB | 60.00s ± 0.00s | 256.00 hours |
| pure bfloat16 | 2 | (1, no, grad_op, sync, paged) | 55.87 GB | 60.09s ± 0.65s | 256.38 hours |
| pure bfloat16 | 2 | (4, yes, grad_op, no_sync, no_paged) | 62.21 GB | 61.00s ± 0.00s | 260.27 hours |
| pure bfloat16 | 2 | (2, no, full, no_sync, paged) | 75.84 GB | 61.00s ± 0.45s | 260.27 hours |
| pure bfloat16 | 2 | (8, yes, grad_op, sync, no_paged) | 65.45 GB | 61.10s ± 0.30s | 260.69 hours |
| pure bfloat16 | 2 | (8, yes, grad_op, no_sync, paged) | 58.21 GB | 61.10s ± 0.30s | 260.69 hours |
| pure bfloat16 | 2 | (8, yes, full, no_sync, no_paged) | 59.61 GB | 61.10s ± 0.30s | 260.69 hours |
| pure bfloat16 | 2 | (8, yes, full, sync, no_paged) | 52.37 GB | 61.50s ± 0.50s | 262.40 hours |
| pure bfloat16 | 2 | (8, yes, full, sync, paged) | 37.87 GB | 61.60s ± 0.49s | 262.83 hours |
| pure bfloat16 | 2 | (4, yes, grad_op, no_sync, paged) | 47.72 GB | 62.30s ± 0.64s | 265.81 hours |
| pure bfloat16 | 2 | (4, yes, grad_op, sync, no_paged) | 54.97 GB | 63.00s ± 0.00s | 268.80 hours |
| pure bfloat16 | 2 | (8, yes, grad_op, sync, paged) | 50.96 GB | 63.00s ± 0.77s | 268.80 hours |
| pure bfloat16 | 2 | (2, yes, grad_op, no_sync, no_paged) | 56.97 GB | 63.10s ± 0.30s | 269.23 hours |
| pure bfloat16 | 2 | (8, yes, full, no_sync, paged) | 45.12 GB | 63.10s ± 0.70s | 269.23 hours |
| pure bfloat16 | 2 | (4, yes, full, no_sync, no_paged) | 49.13 GB | 63.30s ± 0.46s | 270.08 hours |
| pure bfloat16 | 2 | (4, yes, full, sync, paged) | 27.38 GB | 63.60s ± 0.49s | 271.36 hours |
| pure bfloat16 | 2 | (4, yes, full, sync, no_paged) | 41.88 GB | 63.70s ± 0.46s | 271.79 hours |
| pure bfloat16 | 2 | (4, yes, grad_op, sync, paged) | 40.47 GB | 64.70s ± 1.00s | 276.05 hours |
| pure bfloat16 | 2 | (4, yes, full, no_sync, paged) | 34.63 GB | 65.20s ± 0.98s | 278.19 hours |
| pure bfloat16 | 2 | (2, yes, grad_op, no_sync, paged) | 42.48 GB | 65.30s ± 0.90s | 278.61 hours |
| pure bfloat16 | 2 | (2, yes, grad_op, sync, paged) | 35.23 GB | 67.20s ± 0.40s | 286.72 hours |
| pure bfloat16 | 2 | (2, yes, grad_op, sync, no_paged) | 49.72 GB | 67.30s ± 0.46s | 287.15 hours |
| pure bfloat16 | 2 | (1, yes, grad_op, no_sync, no_paged) | 54.35 GB | 67.50s ± 0.50s | 288.00 hours |
| pure bfloat16 | 2 | (2, yes, full, no_sync, paged) | 29.39 GB | 67.90s ± 0.30s | 289.71 hours |
| pure bfloat16 | 2 | (2, yes, full, no_sync, no_paged) | 43.88 GB | 68.00s ± 0.00s | 290.13 hours |
| pure bfloat16 | 2 | (2, yes, full, sync, no_paged) | 37.92 GB | 68.00s ± 0.00s | 290.13 hours |
| pure bfloat16 | 2 | (2, yes, full, sync, paged) | 22.14 GB | 68.00s ± 0.00s | 290.13 hours |
| pure bfloat16 | 2 | (1, yes, grad_op, no_sync, paged) | 39.85 GB | 69.50s ± 1.20s | 296.53 hours |
| pure bfloat16 | 2 | (1, yes, full, no_sync, paged) | 26.77 GB | 73.70s ± 0.46s | 314.45 hours |
| pure bfloat16 | 2 | (1, yes, full, no_sync, no_paged) | 41.26 GB | 74.00s ± 0.00s | 315.73 hours |
| pure bfloat16 | 2 | (1, yes, grad_op, sync, no_paged) | 47.1 GB | 74.00s ± 0.00s | 315.73 hours |
| pure bfloat16 | 2 | (1, yes, grad_op, sync, paged) | 32.6 GB | 74.30s ± 0.46s | 317.01 hours |
| pure bfloat16 | 2 | (1, yes, full, sync, no_paged) | 37.38 GB | 75.00s ± 0.00s | 320.00 hours |
| pure bfloat16 | 2 | (1, yes, full, sync, paged) | 19.52 GB | 75.20s ± 0.40s | 320.85 hours |
| pure bfloat16 | 2 | (2, no, full, sync, no_paged) | OOM | N/A | N/A |
| pure bfloat16 | 2 | (2, no, grad_op, sync, no_paged) | OOM | N/A | N/A |
| pure bfloat16 | 2 | (4, no, full, no_sync, no_paged) | OOM | N/A | N/A |
| pure bfloat16 | 2 | (8, no, grad_op, no_sync, paged) | OOM | N/A | N/A |
| pure bfloat16 | 2 | (2, no, grad_op, no_sync, paged) | OOM | N/A | N/A |
| pure bfloat16 | 2 | (2, no, full, no_sync, no_paged) | OOM | N/A | N/A |
| pure bfloat16 | 2 | (4, no, full, sync, paged) | OOM | N/A | N/A |
| pure bfloat16 | 2 | (4, no, grad_op, no_sync, paged) | OOM | N/A | N/A |
| pure bfloat16 | 2 | (4, no, full, no_sync, paged) | OOM | N/A | N/A |
| pure bfloat16 | 2 | (8, no, grad_op, sync, paged) | OOM | N/A | N/A |
| pure bfloat16 | 2 | (8, no, full, sync, paged) | OOM | N/A | N/A |
| pure bfloat16 | 2 | (2, no, grad_op, sync, paged) | OOM | N/A | N/A |
| pure bfloat16 | 2 | (4, no, grad_op, sync, no_paged) | OOM | N/A | N/A |
| pure bfloat16 | 2 | (8, no, grad_op, sync, no_paged) | OOM | N/A | N/A |
| pure bfloat16 | 2 | (8, no, full, no_sync, paged) | OOM | N/A | N/A |
| pure bfloat16 | 2 | (8, no, grad_op, no_sync, no_paged) | OOM | N/A | N/A |
| pure bfloat16 | 2 | (8, no, full, no_sync, no_paged) | OOM | N/A | N/A |
| pure bfloat16 | 2 | (8, no, full, sync, no_paged) | OOM | N/A | N/A |
| pure bfloat16 | 2 | (2, no, grad_op, no_sync, paged) | OOM | N/A | N/A |
| pure bfloat16 | 2 | (4, no, grad_op, no_sync, no_paged) | OOM | N/A | N/A |
| pure bfloat16 | 2 | (4, no, grad_op, sync, paged) | OOM | N/A | N/A |
| pure bfloat16 | 2 | (4, no, full, no_sync, no_paged) | OOM | N/A | N/A |
| mixed-precision bfloat16 | 4 | (8, yes, full, sync, no_paged) | 64.41 GB | 34.65s ± 0.23s | 295.68 hours |
| mixed-precision bfloat16 | 4 | (8, yes, full, sync, paged) | 49.91 GB | 35.79s ± 0.92s | 305.37 hours |
| mixed-precision bfloat16 | 4 | (4, yes, full, sync, paged) | 34.46 GB | 37.24s ± 0.23s | 317.78 hours |
| mixed-precision bfloat16 | 4 | (4, yes, full, no_sync, no_paged) | 70.7 GB | 37.24s ± 0.24s | 317.81 hours |
| mixed-precision bfloat16 | 4 | (4, yes, full, sync, no_paged) | 48.96 GB | 37.25s ± 0.28s | 317.89 hours |
| mixed-precision bfloat16 | 4 | (4, yes, grad_op, sync, no_paged) | 60.64 GB | 37.71s ± 0.31s | 321.82 hours |
| mixed-precision bfloat16 | 4 | (4, yes, full, no_sync, paged) | 56.21 GB | 38.24s ± 1.36s | 326.31 hours |
| mixed-precision bfloat16 | 4 | (1, no, full, sync, paged) | 63.61 GB | 40.56s ± 0.23s | 346.14 hours |
| mixed-precision bfloat16 | 4 | (2, yes, full, no_sync, no_paged) | 62.98 GB | 41.10s ± 0.14s | 350.72 hours |
| mixed-precision bfloat16 | 4 | (2, yes, full, no_sync, paged) | 52.91 GB | 41.14s ± 0.11s | 351.09 hours |
| mixed-precision bfloat16 | 4 | (2, yes, grad_op, sync, no_paged) | 67.41 GB | 41.15s ± 0.13s | 351.13 hours |
| mixed-precision bfloat16 | 4 | (2, yes, grad_op, no_sync, paged) | 74.65 GB | 41.80s ± 1.00s | 356.67 hours |
| mixed-precision bfloat16 | 4 | (2, yes, full, sync, no_paged) | 41.23 GB | 41.94s ± 0.13s | 357.91 hours |
| mixed-precision bfloat16 | 4 | (2, yes, full, no_sync, paged) | 48.48 GB | 41.97s ± 0.41s | 358.15 hours |
| mixed-precision bfloat16 | 4 | (2, yes, full, sync, paged) | 26.74 GB | 42.10s ± 0.12s | 359.28 hours |
| mixed-precision bfloat16 | 4 | (1, yes, grad_op, no_sync, paged) | 70.8 GB | 45.67s ± 1.26s | 389.73 hours |
| mixed-precision bfloat16 | 4 | (1, yes, full, no_sync, paged) | 44.63 GB | 48.62s ± 0.19s | 414.86 hours |
| mixed-precision bfloat16 | 4 | (1, yes, full, no_sync, no_paged) | 59.13 GB | 48.69s ± 0.16s | 415.45 hours |

| Precision | # GPUs | (mb, ckpt, sharding) | Max. CUDA RAM | Step time | Total GPU Hours |
|---|---|---|---|---|---|
| mixed-precision bfloat16 | 4 | (1, yes, grad_op, sync, no_paged) | 63.55 GB | 48.84s ± 0.17s | 416.81 hours |
| mixed-precision bfloat16 | 4 | (1, yes, full, sync, no_paged) | 37.38 GB | 50.41s ± 0.21s | 430.19 hours |
| mixed-precision bfloat16 | 4 | (1, yes, full, sync, paged) | 22.88 GB | 50.85s ± 0.35s | 433.93 hours |
| mixed-precision bfloat16 | 4 | (1, yes, grad_op, sync, paged) | 49.06 GB | 52.88s ± 0.71s | 451.23 hours |
| mixed-precision bfloat16 | 4 | (1, no, full, sync, no_paged) | 78.1 GB | 81.20s ± 1.08s | 692.91 hours |
| mixed-precision bfloat16 | 4 | (2, no, full, no_sync, paged) | OOM | N/A | N/A |
| mixed-precision bfloat16 | 4 | (4, no, full, no_sync, no_paged) | OOM | N/A | N/A |
| mixed-precision bfloat16 | 4 | (8, no, grad_op, sync, paged) | OOM | N/A | N/A |
| mixed-precision bfloat16 | 4 | (4, yes, grad_op, sync, paged) | OOM | N/A | N/A |
| mixed-precision bfloat16 | 4 | (4, no, grad_op, no_sync, no_paged) | OOM | N/A | N/A |
| mixed-precision bfloat16 | 4 | (4, yes, grad_op, no_sync, no_paged) | OOM | N/A | N/A |
| mixed-precision bfloat16 | 4 | (2, no, grad_op, no_sync, no_paged) | OOM | N/A | N/A |
| mixed-precision bfloat16 | 4 | (8, no, full, sync, paged) | OOM | N/A | N/A |
| mixed-precision bfloat16 | 4 | (8, yes, grad_op, sync, paged) | OOM | N/A | N/A |
| mixed-precision bfloat16 | 4 | (8, yes, full, no_sync, no_paged) | OOM | N/A | N/A |
| mixed-precision bfloat16 | 4 | (1, no, full, no_sync, paged) | OOM | N/A | N/A |
| mixed-precision bfloat16 | 4 | (8, yes, grad_op, no_sync, no_paged) | OOM | N/A | N/A |
| mixed-precision bfloat16 | 4 | (1, yes, grad_op, no_sync, no_paged) | OOM | N/A | N/A |
| mixed-precision bfloat16 | 4 | (8, no, full, no_sync, no_paged) | OOM | N/A | N/A |
| mixed-precision bfloat16 | 4 | (8, no, grad_op, no_sync, no_paged) | OOM | N/A | N/A |
| mixed-precision bfloat16 | 4 | (8, yes, grad_op, sync, no_paged) | OOM | N/A | N/A |
| mixed-precision bfloat16 | 4 | (2, no, full, sync, paged) | OOM | N/A | N/A |
| mixed-precision bfloat16 | 4 | (8, no, full, sync, no_paged) | OOM | N/A | N/A |
| mixed-precision bfloat16 | 4 | (1, no, grad_op, sync, paged) | OOM | N/A | N/A |
| mixed-precision bfloat16 | 4 | (8, yes, grad_op, no_sync, paged) | OOM | N/A | N/A |
| mixed-precision bfloat16 | 4 | (4, no, grad_op, no_sync, paged) | OOM | N/A | N/A |
| mixed-precision bfloat16 | 4 | (1, no, grad_op, no_sync, no_paged) | OOM | N/A | N/A |
| mixed-precision bfloat16 | 4 | (1, no, grad_op, sync, no_paged) | OOM | N/A | N/A |
| mixed-precision bfloat16 | 4 | (8, yes, full, no_sync, paged) | OOM | N/A | N/A |
| mixed-precision bfloat16 | 4 | (4, yes, grad_op, no_sync, paged) | OOM | N/A | N/A |
| mixed-precision bfloat16 | 4 | (1, no, full, no_sync, no_paged) | OOM | N/A | N/A |
| mixed-precision bfloat16 | 4 | (2, no, grad_op, no_sync, paged) | OOM | N/A | N/A |
| mixed-precision bfloat16 | 4 | (8, no, grad_op, sync, no_paged) | OOM | N/A | N/A |
| mixed-precision bfloat16 | 4 | (8, no, grad_op, no_sync, paged) | OOM | N/A | N/A |
| mixed-precision bfloat16 | 4 | (2, no, grad_op, sync, paged) | OOM | N/A | N/A |
| mixed-precision bfloat16 | 4 | (2, no, grad_op, sync, no_paged) | OOM | N/A | N/A |
| mixed-precision bfloat16 | 4 | (4, no, full, no_sync, paged) | OOM | N/A | N/A |
| mixed-precision bfloat16 | 4 | (2, no, full, no_sync, no_paged) | OOM | N/A | N/A |
| mixed-precision bfloat16 | 4 | (4, no, grad_op, sync, paged) | OOM | N/A | N/A |
| mixed-precision bfloat16 | 4 | (1, no, grad_op, no_sync, paged) | OOM | N/A | N/A |
| mixed-precision bfloat16 | 4 | (4, no, full, sync, no_paged) | OOM | N/A | N/A |
| mixed-precision bfloat16 | 4 | (2, yes, grad_op, no_sync, no_paged) | OOM | N/A | N/A |
| mixed-precision bfloat16 | 4 | (8, no, full, no_sync, paged) | OOM | N/A | N/A |
| mixed-precision bfloat16 | 4 | (4, no, full, sync, paged) | OOM | N/A | N/A |
| mixed-precision bfloat16 | 4 | (2, no, full, sync, no_paged) | OOM | N/A | N/A |
| mixed-precision bfloat16 | 4 | (4, no, grad_op, sync, no_paged) | OOM | N/A | N/A |
| pure bfloat16 | 4 | (1, no, grad_op, no_sync, no_paged) | 66.73 GB | 26.46s ± 0.12s | 225.77 hours |
| pure bfloat16 | 4 | (1, no, grad_op, sync, no_paged) | 59.49 GB | 26.57s ± 0.14s | 226.69 hours |
| pure bfloat16 | 4 | (2, no, full, sync, no_paged) | 68.59 GB | 27.19s ± 0.11s | 231.98 hours |
| pure bfloat16 | 4 | (2, no, grad_op, sync, paged) | 74.86 GB | 27.65s ± 0.16s | 235.93 hours |
| pure bfloat16 | 4 | (2, no, full, sync, paged) | 61.34 GB | 28.24s ± 0.34s | 241.01 hours |
| pure bfloat16 | 4 | (2, no, full, no_sync, paged) | 72.21 GB | 28.57s ± 1.53s | 243.83 hours |
| pure bfloat16 | 4 | (1, no, full, no_sync, no_paged) | 53.43 GB | 29.91s ± 0.24s | 255.27 hours |
| pure bfloat16 | 4 | (1, no, full, no_sync, paged) | 46.18 GB | 29.94s ± 0.25s | 255.49 hours |
| pure bfloat16 | 4 | (1, no, grad_op, sync, no_paged) | 55.86 GB | 29.99s ± 0.24s | 255.94 hours |
| pure bfloat16 | 4 | (1, no, grad_op, sync, paged) | 48.61 GB | 30.10s ± 0.21s | 256.84 hours |
| pure bfloat16 | 4 | (8, yes, grad_op, no_sync, no_paged) | 61.82 GB | 30.36s ± 0.19s | 259.10 hours |
| pure bfloat16 | 4 | (4, yes, grad_op, no_sync, paged) | 44.09 GB | 30.79s ± 0.10s | 262.74 hours |
| pure bfloat16 | 4 | (4, yes, grad_op, no_sync, no_paged) | 51.34 GB | 30.81s ± 0.10s | 262.95 hours |
| pure bfloat16 | 4 | (1, no, full, sync, no_paged) | 42.56 GB | 30.85s ± 0.18s | 263.29 hours |
| pure bfloat16 | 4 | (1, no, full, sync, paged) | 35.31 GB | 31.02s ± 0.23s | 264.69 hours |
| pure bfloat16 | 4 | (8, yes, grad_op, sync, no_paged) | 50.95 GB | 31.10s ± 0.20s | 265.39 hours |
| pure bfloat16 | 4 | (8, yes, full, no_sync, paged) | 41.49 GB | 31.20s ± 0.23s | 266.21 hours |
| pure bfloat16 | 4 | (8, yes, full, no_sync, no_paged) | 48.74 GB | 31.20s ± 0.20s | 266.22 hours |
| pure bfloat16 | 4 | (8, yes, grad_op, sync, paged) | 43.71 GB | 31.20s ± 0.24s | 266.23 hours |
| pure bfloat16 | 4 | (8, yes, full, sync, no_paged) | 37.87 GB | 31.26s ± 0.21s | 266.74 hours |
| pure bfloat16 | 4 | (8, yes, full, sync, paged) | 30.62 GB | 31.32s ± 0.26s | 267.27 hours |
| pure bfloat16 | 4 | (8, yes, grad_op, no_sync, paged) | 54.58 GB | 31.42s ± 0.53s | 268.14 hours |
| pure bfloat16 | 4 | (2, yes, grad_op, no_sync, no_paged) | 46.1 GB | 32.02s ± 0.09s | 273.22 hours |
| pure bfloat16 | 4 | (2, yes, grad_op, no_sync, paged) | 38.85 GB | 32.11s ± 0.07s | 274.05 hours |
| pure bfloat16 | 4 | (4, yes, grad_op, sync, paged) | 33.22 GB | 32.38s ± 0.24s | 276.31 hours |
| pure bfloat16 | 4 | (4, yes, grad_op, sync, no_paged) | 40.47 GB | 32.39s ± 0.27s | 276.38 hours |
| pure bfloat16 | 4 | (4, yes, full, no_sync, no_paged) | 27.38 GB | 32.58s ± 0.25s | 277.99 hours |
| pure bfloat16 | 4 | (4, yes, full, no_sync, paged) | 31.01 GB | 32.58s ± 0.24s | 278.00 hours |
| pure bfloat16 | 4 | (4, yes, full, no_sync, no_paged) | 38.25 GB | 32.62s ± 0.27s | 278.32 hours |
| pure bfloat16 | 4 | (4, yes, full, sync, paged) | 20.13 GB | 32.62s ± 0.25s | 278.36 hours |
| pure bfloat16 | 4 | (1, yes, grad_op, no_sync, no_paged) | 43.48 GB | 34.20s ± 0.13s | 291.88 hours |

| Precision | # GPUs | (mb, ckpt, sharding) | Max. CUDA RAM | Step time | Total GPU Hours |
|---|---|---|---|---|---|
| pure bfloat16 | 4 | (1, yes, grad_op, no_sync, paged) | 36.23 GB | 34.36s ± 0.24s | 293.17 hours |
| pure bfloat16 | 4 | (2, yes, grad_op, sync, no_paged) | 35.23 GB | 34.85s ± 0.14s | 297.39 hours |
| pure bfloat16 | 4 | (2, yes, grad_op, sync, paged) | 27.98 GB | 34.85s ± 0.13s | 297.42 hours |
| pure bfloat16 | 4 | (2, yes, full, no_sync, no_paged) | 33.01 GB | 35.11s ± 0.12s | 299.56 hours |
| pure bfloat16 | 4 | (2, yes, full, no_sync, no_paged) | 25.76 GB | 35.18s ± 0.13s | 300.25 hours |
| pure bfloat16 | 4 | (2, yes, full, sync, no_paged) | 22.14 GB | 35.25s ± 0.14s | 300.78 hours |
| pure bfloat16 | 4 | (2, yes, full, sync, paged) | 14.89 GB | 35.30s ± 0.13s | 301.21 hours |
| pure bfloat16 | 4 | (1, yes, full, no_sync, no_paged) | 30.39 GB | 38.52s ± 0.16s | 328.66 hours |
| pure bfloat16 | 4 | (1, yes, full, no_sync, paged) | 23.14 GB | 38.57s ± 0.27s | 329.15 hours |
| pure bfloat16 | 4 | (1, yes, grad_op, sync, no_paged) | 32.6 GB | 38.88s ± 0.18s | 331.79 hours |
| pure bfloat16 | 4 | (1, yes, grad_op, sync, paged) | 25.36 GB | 38.92s ± 0.18s | 332.13 hours |
| pure bfloat16 | 4 | (1, yes, full, sync, no_paged) | 19.52 GB | 39.52s ± 0.19s | 337.26 hours |
| pure bfloat16 | 4 | (1, yes, full, sync, paged) | 12.27 GB | 39.97s ± 0.38s | 341.11 hours |
| pure bfloat16 | 4 | (4, no, full, sync, no_paged) | OOM | N/A | N/A |
| pure bfloat16 | 4 | (2, no, grad_op, no_sync, paged) | OOM | N/A | N/A |
| pure bfloat16 | 4 | (2, no, full, no_sync, no_paged) | OOM | N/A | N/A |
| pure bfloat16 | 4 | (4, no, full, no_sync, no_paged) | OOM | N/A | N/A |
| pure bfloat16 | 4 | (4, no, grad_op, no_sync, paged) | OOM | N/A | N/A |
| pure bfloat16 | 4 | (8, no, full, sync, no_paged) | OOM | N/A | N/A |
| pure bfloat16 | 4 | (4, no, grad_op, no_sync, no_paged) | OOM | N/A | N/A |
| pure bfloat16 | 4 | (2, no, grad_op, sync, no_paged) | OOM | N/A | N/A |
| pure bfloat16 | 4 | (8, no, full, sync, paged) | OOM | N/A | N/A |
| pure bfloat16 | 4 | (2, no, grad_op, no_sync, no_paged) | OOM | N/A | N/A |
| pure bfloat16 | 4 | (4, no, grad_op, sync, no_paged) | OOM | N/A | N/A |
| pure bfloat16 | 4 | (4, no, full, no_sync, no_paged) | OOM | N/A | N/A |
| pure bfloat16 | 4 | (8, no, grad_op, no_sync, no_paged) | OOM | N/A | N/A |
| pure bfloat16 | 4 | (8, no, grad_op, no_sync, paged) | OOM | N/A | N/A |
| pure bfloat16 | 4 | (4, no, grad_op, sync, paged) | OOM | N/A | N/A |
| pure bfloat16 | 4 | (8, no, grad_op, sync, no_paged) | OOM | N/A | N/A |
| pure bfloat16 | 4 | (8, no, full, no_sync, no_paged) | OOM | N/A | N/A |
| pure bfloat16 | 4 | (4, no, full, sync, paged) | OOM | N/A | N/A |
| pure bfloat16 | 4 | (8, no, full, no_sync, no_paged) | OOM | N/A | N/A |
| pure bfloat16 | 4 | (8, no, grad_op, sync, paged) | OOM | N/A | N/A |
| mixed-precision bfloat16 | 8 | (8, yes, full, sync, paged) | 42.66 GB | 17.46s ± 0.20s | 298.00 hours |
| mixed-precision bfloat16 | 8 | (8, yes, full, sync, no_paged) | 49.91 GB | 17.48s ± 0.23s | 298.33 hours |
| mixed-precision bfloat16 | 8 | (4, yes, grad_op, sync, no_paged) | 60.63 GB | 18.53s ± 0.22s | 316.25 hours |
| mixed-precision bfloat16 | 8 | (4, yes, full, no_sync, no_paged) | 59.83 GB | 18.66s ± 0.20s | 318.50 hours |
| mixed-precision bfloat16 | 8 | (4, yes, full, no_sync, paged) | 52.58 GB | 18.69s ± 0.25s | 319.03 hours |
| mixed-precision bfloat16 | 8 | (4, yes, full, sync, no_paged) | 34.46 GB | 18.75s ± 0.23s | 319.97 hours |
| mixed-precision bfloat16 | 8 | (4, yes, full, sync, paged) | 27.21 GB | 18.86s ± 0.34s | 321.89 hours |
| mixed-precision bfloat16 | 8 | (4, yes, grad_op, sync, paged) | 53.39 GB | 19.17s ± 0.30s | 327.13 hours |
| mixed-precision bfloat16 | 8 | (1, no, full, sync, no_paged) | 63.6 GB | 20.06s ± 0.11s | 342.34 hours |
| mixed-precision bfloat16 | 8 | (2, yes, grad_op, no_sync, paged) | 71.03 GB | 20.18s ± 3.06s | 344.46 hours |
| mixed-precision bfloat16 | 8 | (1, no, full, sync, paged) | 56.35 GB | 20.57s ± 0.21s | 350.99 hours |
| mixed-precision bfloat16 | 8 | (2, yes, grad_op, sync, no_paged) | 52.91 GB | 20.83s ± 0.26s | 355.48 hours |
| mixed-precision bfloat16 | 8 | (2, yes, full, no_sync, no_paged) | 52.1 GB | 21.01s ± 0.28s | 358.62 hours |
| mixed-precision bfloat16 | 8 | (2, yes, full, sync, no_paged) | 26.74 GB | 21.23s ± 0.25s | 362.39 hours |
| mixed-precision bfloat16 | 8 | (2, yes, full, sync, paged) | 19.49 GB | 21.36s ± 0.36s | 364.63 hours |
| mixed-precision bfloat16 | 8 | (2, yes, grad_op, sync, paged) | 45.66 GB | 21.49s ± 0.32s | 366.73 hours |
| mixed-precision bfloat16 | 8 | (1, yes, grad_op, no_sync, no_paged) | 74.43 GB | 21.52s ± 0.14s | 367.34 hours |
| mixed-precision bfloat16 | 8 | (2, yes, full, no_sync, paged) | 44.86 GB | 21.72s ± 0.38s | 370.65 hours |
| mixed-precision bfloat16 | 8 | (1, yes, grad_op, no_sync, paged) | 67.18 GB | 21.73s ± 0.05s | 370.82 hours |
| mixed-precision bfloat16 | 8 | (1, yes, grad_op, sync, no_paged) | 49.06 GB | 24.77s ± 0.11s | 422.83 hours |
| mixed-precision bfloat16 | 8 | (1, yes, full, no_sync, no_paged) | 48.25 GB | 25.01s ± 0.09s | 426.91 hours |
| mixed-precision bfloat16 | 8 | (1, yes, full, no_sync, paged) | 41.01 GB | 25.66s ± 0.23s | 437.90 hours |
| mixed-precision bfloat16 | 8 | (1, yes, full, sync, no_paged) | 22.88 GB | 25.75s ± 0.10s | 439.40 hours |
| mixed-precision bfloat16 | 8 | (1, yes, grad_op, sync, paged) | 41.81 GB | 26.16s ± 0.36s | 446.38 hours |
| mixed-precision bfloat16 | 8 | (1, yes, full, sync, paged) | 15.64 GB | 26.18s ± 0.17s | 446.82 hours |
| mixed-precision bfloat16 | 8 | (4, no, full, sync, no_paged) | OOM | N/A | N/A |
| mixed-precision bfloat16 | 8 | (8, yes, grad_op, no_sync, no_paged) | OOM | N/A | N/A |
| mixed-precision bfloat16 | 8 | (2, no, grad_op, no_sync, no_paged) | OOM | N/A | N/A |
| mixed-precision bfloat16 | 8 | (8, yes, full, no_sync, paged) | OOM | N/A | N/A |
| mixed-precision bfloat16 | 8 | (8, no, grad_op, no_sync, paged) | OOM | N/A | N/A |
| mixed-precision bfloat16 | 8 | (2, no, grad_op, no_sync, paged) | OOM | N/A | N/A |
| mixed-precision bfloat16 | 8 | (1, no, grad_op, sync, no_paged) | OOM | N/A | N/A |
| mixed-precision bfloat16 | 8 | (4, no, grad_op, sync, no_paged) | OOM | N/A | N/A |
| mixed-precision bfloat16 | 8 | (2, yes, grad_op, no_sync, no_paged) | OOM | N/A | N/A |
| mixed-precision bfloat16 | 8 | (8, no, grad_op, sync, paged) | OOM | N/A | N/A |
| mixed-precision bfloat16 | 8 | (8, no, full, sync, paged) | OOM | N/A | N/A |
| mixed-precision bfloat16 | 8 | (4, no, full, no_sync, no_paged) | OOM | N/A | N/A |
| mixed-precision bfloat16 | 8 | (8, no, full, no_sync, no_paged) | OOM | N/A | N/A |
| mixed-precision bfloat16 | 8 | (1, no, full, no_sync, no_paged) | OOM | N/A | N/A |
| mixed-precision bfloat16 | 8 | (8, yes, grad_op, no_sync, paged) | OOM | N/A | N/A |
| mixed-precision bfloat16 | 8 | (2, no, full, no_sync, paged) | OOM | N/A | N/A |
| mixed-precision bfloat16 | 8 | (8, no, grad_op, sync, no_paged) | OOM | N/A | N/A |
| mixed-precision bfloat16 | 8 | (4, no, grad_op, sync, paged) | OOM | N/A | N/A |
| mixed-precision bfloat16 | 8 | (4, yes, grad_op, no_sync, paged) | OOM | N/A | N/A |

| Precision | # GPUs | (mb, ckpt, sharding) | Max. CUDA RAM | Step time | Total GPU Hours |
|---|---|---|---|---|---|
| mixed-precision bfloat16 | 8 | (8, no, full, sync, no_paged) | OOM | N/A | N/A |
| mixed-precision bfloat16 | 8 | (8, no, full, no_sync, paged) | OOM | N/A | N/A |
| mixed-precision bfloat16 | 8 | (8, yes, grad_op, sync, paged) | OOM | N/A | N/A |
| mixed-precision bfloat16 | 8 | (4, no, grad_op, no_sync, no_paged) | OOM | N/A | N/A |
| mixed-precision bfloat16 | 8 | (1, no, grad_op, no_sync, paged) | OOM | N/A | N/A |
| mixed-precision bfloat16 | 8 | (8, yes, full, no_sync, no_paged) | OOM | N/A | N/A |
| mixed-precision bfloat16 | 8 | (4, no, grad_op, no_sync, paged) | OOM | N/A | N/A |
| mixed-precision bfloat16 | 8 | (2, no, grad_op, sync, paged) | OOM | N/A | N/A |
| mixed-precision bfloat16 | 8 | (2, no, full, sync, no_paged) | OOM | N/A | N/A |
| mixed-precision bfloat16 | 8 | (2, no, full, sync, paged) | OOM | N/A | N/A |
| mixed-precision bfloat16 | 8 | (4, no, full, sync, paged) | OOM | N/A | N/A |
| mixed-precision bfloat16 | 8 | (1, no, full, no_sync, paged) | OOM | N/A | N/A |
| mixed-precision bfloat16 | 8 | (1, no, grad_op, sync, paged) | OOM | N/A | N/A |
| mixed-precision bfloat16 | 8 | (1, no, grad_op, no_sync, no_paged) | OOM | N/A | N/A |
| mixed-precision bfloat16 | 8 | (2, no, full, no_sync, no_paged) | OOM | N/A | N/A |
| mixed-precision bfloat16 | 8 | (8, no, grad_op, no_sync, no_paged) | OOM | N/A | N/A |
| mixed-precision bfloat16 | 8 | (4, yes, grad_op, no_sync, no_paged) | OOM | N/A | N/A |
| mixed-precision bfloat16 | 8 | (8, yes, grad_op, sync, no_paged) | OOM | N/A | N/A |
| mixed-precision bfloat16 | 8 | (4, no, full, no_sync, paged) | OOM | N/A | N/A |
| mixed-precision bfloat16 | 8 | (2, no, grad_op, sync, no_paged) | OOM | N/A | N/A |
| pure bfloat16 | 8 | (1, no, grad_op, no_sync, no_paged) | 61.29 GB | 13.45s ± 0.07s | 229.55 hours |
| pure bfloat16 | 8 | (1, no, grad_op, no_sync, paged) | 57.67 GB | 13.55s ± 0.12s | 231.25 hours |
| pure bfloat16 | 8 | (2, no, grad_op, no_sync, paged) | 74.85 GB | 13.65s ± 0.24s | 232.87 hours |
| pure bfloat16 | 8 | (2, no, full, sync, no_paged) | 61.33 GB | 13.78s ± 0.10s | 235.26 hours |
| pure bfloat16 | 8 | (2, no, full, sync, paged) | 57.71 GB | 13.80s ± 0.10s | 235.55 hours |
| pure bfloat16 | 8 | (2, no, full, no_sync, no_paged) | 74.01 GB | 14.19s ± 0.19s | 242.21 hours |
| pure bfloat16 | 8 | (2, no, grad_op, sync, paged) | 71.23 GB | 14.35s ± 0.31s | 244.92 hours |
| pure bfloat16 | 8 | (2, no, full, no_sync, paged) | 70.39 GB | 14.58s ± 0.31s | 248.81 hours |
| pure bfloat16 | 8 | (1, no, grad_op, sync, no_paged) | 48.61 GB | 15.28s ± 0.08s | 260.78 hours |
| pure bfloat16 | 8 | (8, yes, grad_op, no_sync, paged) | 52.76 GB | 15.30s ± 0.13s | 261.15 hours |
| pure bfloat16 | 8 | (8, yes, grad_op, no_sync, no_paged) | 56.39 GB | 15.33s ± 0.17s | 261.70 hours |
| pure bfloat16 | 8 | (1, no, grad_op, sync, paged) | 44.99 GB | 15.38s ± 0.15s | 262.47 hours |
| pure bfloat16 | 8 | (1, no, full, no_sync, no_paged) | 47.99 GB | 15.46s ± 0.08s | 263.87 hours |
| pure bfloat16 | 8 | (1, no, full, no_sync, paged) | 44.37 GB | 15.59s ± 0.14s | 266.04 hours |
| pure bfloat16 | 8 | (4, yes, grad_op, no_sync, no_paged) | 45.9 GB | 15.59s ± 0.16s | 266.15 hours |
| pure bfloat16 | 8 | (4, yes, grad_op, no_sync, paged) | 42.28 GB | 15.60s ± 0.17s | 266.22 hours |
| pure bfloat16 | 8 | (8, yes, grad_op, sync, paged) | 40.08 GB | 15.72s ± 0.22s | 268.25 hours |
| pure bfloat16 | 8 | (8, yes, full, no_sync, no_paged) | 43.3 GB | 15.74s ± 0.16s | 268.65 hours |
| pure bfloat16 | 8 | (8, yes, grad_op, sync, no_paged) | 43.7 GB | 15.74s ± 0.30s | 268.70 hours |
| pure bfloat16 | 8 | (8, yes, full, no_sync, paged) | 39.68 GB | 15.77s ± 0.22s | 269.16 hours |
| pure bfloat16 | 8 | (8, yes, full, sync, no_paged) | 30.62 GB | 15.79s ± 0.24s | 269.57 hours |
| pure bfloat16 | 8 | (8, yes, full, sync, paged) | 26.99 GB | 15.81s ± 0.25s | 269.81 hours |
| pure bfloat16 | 8 | (1, no, full, sync, no_paged) | 35.3 GB | 15.82s ± 0.09s | 269.94 hours |
| pure bfloat16 | 8 | (1, no, full, sync, paged) | 31.68 GB | 15.90s ± 0.13s | 271.38 hours |
| pure bfloat16 | 8 | (2, yes, grad_op, no_sync, no_paged) | 40.66 GB | 16.13s ± 0.10s | 275.29 hours |
| pure bfloat16 | 8 | (2, yes, grad_op, no_sync, paged) | 37.04 GB | 16.13s ± 0.12s | 275.32 hours |
| pure bfloat16 | 8 | (4, yes, grad_op, sync, no_paged) | 33.22 GB | 16.33s ± 0.20s | 278.72 hours |
| pure bfloat16 | 8 | (4, yes, full, no_sync, no_paged) | 32.82 GB | 16.40s ± 0.21s | 279.93 hours |
| pure bfloat16 | 8 | (4, yes, grad_op, sync, paged) | 29.6 GB | 16.41s ± 0.38s | 280.15 hours |
| pure bfloat16 | 8 | (4, yes, full, sync, no_paged) | 20.13 GB | 16.44s ± 0.28s | 280.56 hours |
| pure bfloat16 | 8 | (4, yes, full, no_sync, paged) | 29.19 GB | 16.46s ± 0.23s | 280.93 hours |
| pure bfloat16 | 8 | (4, yes, full, sync, paged) | 16.51 GB | 16.49s ± 0.28s | 281.46 hours |
| pure bfloat16 | 8 | (1, yes, grad_op, no_sync, paged) | 34.42 GB | 17.33s ± 0.05s | 295.71 hours |
| pure bfloat16 | 8 | (1, yes, grad_op, no_sync, no_paged) | 38.04 GB | 17.34s ± 0.09s | 295.88 hours |
| pure bfloat16 | 8 | (2, yes, grad_op, sync, no_paged) | 27.98 GB | 17.65s ± 0.26s | 301.28 hours |
| pure bfloat16 | 8 | (2, yes, grad_op, sync, paged) | 24.35 GB | 17.80s ± 0.37s | 303.87 hours |
| pure bfloat16 | 8 | (2, yes, full, no_sync, no_paged) | 27.58 GB | 17.88s ± 0.28s | 305.22 hours |
| pure bfloat16 | 8 | (2, yes, full, no_sync, paged) | 23.95 GB | 17.92s ± 0.34s | 305.78 hours |
| pure bfloat16 | 8 | (2, yes, full, sync, no_paged) | 14.89 GB | 17.93s ± 0.25s | 306.01 hours |
| pure bfloat16 | 8 | (2, yes, full, sync, paged) | 11.27 GB | 18.07s ± 0.39s | 308.43 hours |
| pure bfloat16 | 8 | (1, yes, full, no_sync, no_paged) | 24.95 GB | 19.74s ± 0.12s | 336.86 hours |
| pure bfloat16 | 8 | (1, yes, full, no_sync, paged) | 21.33 GB | 19.84s ± 0.18s | 338.55 hours |
| pure bfloat16 | 8 | (1, yes, grad_op, sync, paged) | 21.73 GB | 20.22s ± 0.21s | 345.12 hours |
| pure bfloat16 | 8 | (1, yes, grad_op, sync, no_paged) | 25.36 GB | 20.28s ± 0.27s | 346.16 hours |
| pure bfloat16 | 8 | (1, yes, full, sync, no_paged) | 12.27 GB | 20.30s ± 0.11s | 346.49 hours |
| pure bfloat16 | 8 | (1, yes, full, sync, paged) | 10.48 GB | 20.74s ± 0.18s | 353.98 hours |
| pure bfloat16 | 8 | (8, no, full, no_sync, paged) | OOM | N/A | N/A |
| pure bfloat16 | 8 | (2, no, grad_op, no_sync, paged) | OOM | N/A | N/A |
| pure bfloat16 | 8 | (4, no, full, no_sync, no_paged) | OOM | N/A | N/A |
| pure bfloat16 | 8 | (4, no, full, sync, no_paged) | OOM | N/A | N/A |
| pure bfloat16 | 8 | (4, no, grad_op, sync, paged) | OOM | N/A | N/A |
| pure bfloat16 | 8 | (8, no, full, sync, no_paged) | OOM | N/A | N/A |
| pure bfloat16 | 8 | (8, no, full, no_sync, no_paged) | OOM | N/A | N/A |
| pure bfloat16 | 8 | (4, no, full, no_sync, paged) | OOM | N/A | N/A |
| pure bfloat16 | 8 | (4, no, grad_op, no_sync, paged) | OOM | N/A | N/A |
| pure bfloat16 | 8 | (4, no, grad_op, no_sync, no_paged) | OOM | N/A | N/A |
| pure bfloat16 | 8 | (8, no, grad_op, no_sync, paged) | OOM | N/A | N/A |

| Precision | # GPUs | (mb, ckpt, sharding) | Max. CUDA RAM | Step time | Total GPU Hours |
|---|---|---|---|---|---|
| pure bfloat16 | 8 | (8, no, grad_op, sync, no_paged) | OOM | N/A | N/A |
| pure bfloat16 | 8 | (8, no, grad_op, sync, paged) | OOM | N/A | N/A |
| pure bfloat16 | 8 | (8, no, full, sync, paged) | OOM | N/A | N/A |
| pure bfloat16 | 8 | (4, no, full, sync, paged) | OOM | N/A | N/A |
| pure bfloat16 | 8 | (4, no, grad_op, sync, no_paged) | OOM | N/A | N/A |
| pure bfloat16 | 8 | (8, no, grad_op, no_sync, no_paged) | OOM | N/A | N/A |
| pure bfloat16 | 8 | (2, no, grad_op, no_sync, no_paged) | OOM | N/A | N/A |

