# OpenReview forum: "Language Adaptation on a Tight Academic Compute Budget: Tokenizer Swapping Works and Pure bfloat16 Is Enough"
_ICML.cc/2024/Workshop/WANT — WANT@ICML 2024 Poster_

### Official Review · Reviewer_4wL9 · 2024-06-12
**This paper is somewhat insightful in using bf16 for continued-pretraining**

**Confidence:** 4

**Summary:**

Interesting results with good analysis. Paper writing can be improved and larger scale experiments can make the results even more convincing.

**Strengths:**

Overall good quality of the analysis and experiments. Analysis of the regularization effect of bf16 optimization is interesting.

It is quite interesting to see that the regularization effect of using a lower precision update would cause the layer-based difference.
I am convinced that using bf16 update leads to a similar effect as layer-frozen (or adapted learning rate), even though it is not clear whether its effect on other layers is also important to the performance.

**Weaknesses:**

It would be nice to see the performance if  RMSNorm layers are frozen over the entire training session.

Page 2 line 55 "mixed-precision bfloat16 training will run out of memory or is only possible if used with inefficient memory-saving techniques like activation checkpointing". There are multiple solutions to reduce the memory peak (for example, the paged optimizer from QLoRA) and they may not cause too much time overhead as gradient checkpointing.
If I understand correctly, the speed-up of using pure bf-16 is mostly coming from not applying the gradient checkpointing. Other memory-saving techniques should be compared to reach a more convincing conclusion.

There are multiple compiled errors with ? in Page 4 line 205, Page 6 lines 294-296, Page 8 line 421

---

### Official Review · Reviewer_Ac9u · 2024-06-14
**Interesting work but needs more experimental evidence to make conclusions more robust**

**Confidence:** 4

**Summary:**

The authors investigated methods for adapting LLMs to different languages under constrained computational resources. The main focus is on continuing the pretraining of the Mistral-7B model for German and Arabic, evaluating techniques to enhance efficiency (swapping tokenizer, and using pure bfloat16 precision training) and performance when only a few GPUs are available.

The main contributions and findings are as follows -
1. Training Precision - Pure bfloat16 training is a viable alternative to mixed-precision training, offering substantial efficiency gains without significant performance loss, particularly beneficial when limited to fewer GPUs.
2. Tokenizer Swapping: Replacing the original tokenizer with a specialized one is effective, providing efficient tokenization and maintaining performance levels.
3. Adapting Mistral-7B to German underperformed compared to the base model, while adaptation to Arabic showed significant improvement, indicating that adaptation is more beneficial for less well-represented languages.

**Strengths:**

- The research addresses critical issues for academic setting on how to do continued pre-training for LLMs in resource constrained environments ( with availability of few GPUs)

- The paper provides good evidence for mixed precision and bfloat16 training and provides reasonable explanation for the outcomes.

- The investigation into tokenizer swapping offers a novel perspective on improving tokenization efficiency, which could be broadly applicable in LLM adaptations.

- The paper is well-structured, with a clear presentation of the problem, methodology, results, and conclusions.

**Weaknesses:**

- The definition of a "tight academic compute budget" is somewhat specific, focusing on server-grade GPUs like Nvidia A100s. This might not fully represent the variability in computational resources available across different academic institutions.

- The finding that adaptation to well-represented languages (like German) may not always yield benefits is significant but needs further exploration to understand the underlying reasons and potential solutions.

- The reduced number of training steps and tokens compared to the reference LeoLM project might limit the depth of the analysis. Exploring longer training durations, even within constrained budgets, could provide more comprehensive insights.

**Limitations:**

- The experiments are only performed on Mixtral-7B which lacks generalizability. There is a need to perform benchmarking on other open source models as well.

- The experiments are conducted on only two languages, German and Arabic. While these languages provide valuable insights, a broader range of languages would strengthen the generalizability of the conclusions.

---

### Decision · Program_Chairs · 2024-06-18

**Decision:**

Accept (Poster)

**Comment:**

We thank the authors for their time and contribution to WANT and we are pleased to share that after the reviewing process the paper has been accepted. Congratulations! We encourage the authors to consider reviewers' feedback for the improvement of the camera-ready version. We hope to see you in person at the workshop and brainstorm on efficient training research together!